# MECHANISM DESIGN WITH MULTI-ARMED BANDIT

## ABSTRACT

A popular approach of automated mechanism design is to formulate a linear program (LP) whose solution gives a mechanism with desired properties. We analytically derive a class of optimal solutions for such an LP that gives mechanisms achieving standard properties of efficiency, incentive compatibility, strong budget balance (SBB), and individual rationality (IR), where SBB and IR are satisfied in expectation. Notably, our solutions are represented by an exponentially smaller number of essential variables than the original variables of LP. Our solutions, however, involve a term whose exact evaluation requires solving a certain optimization problem exponentially many times as the number of players, $N$, grows. We thus evaluate this term by modeling it as the problem of estimating the mean reward of the best arm in multi-armed bandit (MAB), propose a Probably and Approximately Correct estimator, and prove its asymptotic optimality by establishing a lower bound on its sample complexity. This MAB approach reduces the number of times the optimization problem is solved from exponential to $O(N \log N)$. Numerical experiments show that the proposed approach finds mechanisms that are guaranteed to achieve desired properties with high probability for environments with up to 128 players, which substantially improves upon the prior work.

## 1 INTRODUCTION

Multi-agent systems can be made efficient by mediators who make system-wide (social) decisions in a way that maximizes social welfare. For example, in a trading network where firms sell goods to each other or to external markets (Hatfield et al., 2013), the mediator can ensure that those traded goods are produced by the firms with the lowest costs and purchased by those with the highest needs (Osogami et al., 2023). While such mediators could maximize their own profit by charging participants (e.g., today's tech giants who operate consumer marketplaces or ads platforms), they would typically take most of the profit, leaving the participants with only a small portion.

We instead envision an open platform whose purpose is to provide maximal benefits to the participants in multi-agent systems. This is similar to the purpose of designing an auction mechanism for *public* resources, which should be given to those who need them most, and there should be neither budget deficits nor surpluses on the mediator (Bailey, 1997; Cavallo, 2006; Dufton et al., 2021; Gujar & Narahari, 2011; Guo, 2012; Guo & Conitzer, 2009; Manisha et al., 2018; Tacchetti et al., 2022). However, such mechanisms exploit the particular structure of single-sided auctions, where all participants are buyers, and it may be impossible to achieve desired properties in other multi-agent systems such as double-sided auctions (Hobbs et al., 2000; Zou, 2009; Widmer & Leukel, 2016; Stößer et al., 2010; Kumar et al., 2018; Chichin et al., 2017), matching markets (Zhang & Zhu, 2020), and trading networks (Osogami et al., 2023; Wasserkrug & Osogami, 2023) even if those properties could be guaranteed for single-sided auctions.

Here, we design mechanisms for general environments that include all these multi-agent systems, with the primary objectives of efficiency and strong budget balance (SBB). Specifically, we require that the mediator chooses a social decision such that its total value to the players (or participants) is maximized (decision efficiency; DE) and that the expected revenue of the mediator is equal to a target value, $\rho \in \mathbb{R}$ (SBB when $\rho = 0$). As is standard in Bayesian mechanism design (Shoham & Leyton-Brown, 2009), we assume that the value of each social decision to a player depends on the player's type that is only known to that player, but a joint probability distribution of the players' types is a common knowledge. Since it would be difficult to achieve DE without knowing the types, we also require that the optimal strategy of each player is to truthfully declare its type regardless of the

strategies of the others (dominant strategy incentive compatibility; DSIC). To promote participation, we also require that the expected utility of each player is no less than a target value, $\theta(t_n) \in \mathbb{R}$, that can depend on the type $t_n$ of the player (individual rationality or IR when $\theta \equiv 0$).

Although these properties are standard in the literature of mechanism design (Shoham & Leyton-Brown, 2009), here we have introduced parameters, $\rho$ and $\theta$, to generalize the standard definitions of SBB ($\rho = 0$) and IR ($\theta \equiv 0$) with three motivations. First, it is not always possible to achieve the four desired properties with the standard definitions of IR and SBB (Green & Laffont, 1977; Myerson & Satterthwaite, 1983; Osogami et al., 2023), while the generalization will enable the exact characterization of when those properties can be satisfied. Second, this generalization will allow us to develop a principled approach of *learning* a mechanism, where some of the quantities are estimated from samples, *with theoretical guarantees*. The mechanism designed with our learning approach can be shown to satisfy the four desired properties with high probability. Finally, the additional parameters will allow us to model practical requirements. For example, the mediator might need positive revenue to cover the cost of maintaining the platform or might want to guarantee positive expected utility to players to encourage participation in this platform rather than others.

We require SBB and IR *in expectation* (*ex ante* or interim) with respect to the distribution of types, while DE and DSIC are satisfied for any realization of types (*ex post*). While these assumptions are similar to those in Osogami et al. (2023); Wasserkrug & Osogami (2023) for trading networks, the *in expectation* properties are certainly weaker than the *ex post* properties usually assumed for auctions (Bailey, 1997; Cavallo, 2006; Dufton et al., 2021; Gujar & Narahari, 2011; Guo, 2012; Guo & Conitzer, 2009; Manisha et al., 2018; Tacchetti et al., 2022). With the weaker properties, however, we derive *analytical* solutions of the mechanisms that satisfy all the four desired properties in the general environments (and characterize when those properties can be satisfied). This is in stark contrast to the prior work, where mechanisms are analytically derived only for auctions with a single type of goods (Bailey, 1997; Cavallo, 2006; Guo, 2011; Guo & Conitzer, 2007; 2009; Moulin, 2009) or with unit demand (Gujar & Narahari, 2011; Guo, 2012). For more complex auctions (Dufton et al., 2021; Manisha et al., 2018; Tacchetti et al., 2022) or trading networks (Osogami et al., 2023; Wasserkrug & Osogami, 2023), mechanisms are computed by numerically solving optimization problems, whose size often grows exponentially with the number of players.

While the numerical approaches proposed in Osogami et al. (2023) have been applied only to the trading networks with two players, our analytical solutions can be evaluated numerically with $\sim 10$ players, depending on the number of types. The key bottleneck in our analytical solutions lies in the evaluation of the minimum expected value over possible types of each player. Exact evaluation of this quantity would require computing an efficient social decision for all the $K^N$ combinations of types, where $K$ is the number of possible types of each player, and $N$ is the number of players.

To overcome this bottleneck, we model the problem of evaluating this minimum expected value in a multi-armed bandit (MAB) approach (Lattimore & Szepesvári, 2020) and propose an asymptotically optimal learning algorithm for this problem. While the standard objectives of MAB are regret minimization (Auer et al., 1995; 2002) and best arm identification (Audibert et al., 2010; Maron & Moore, 1993; Mnih et al., 2008; Bubeck et al., 2009), our objective is to estimate the mean reward of the best arm. We propose a probably approximately correct (PAC) algorithm, which approximately (with error at most $\varepsilon$) estimates the best mean with high probability (at least $1 - \delta$), and proves that its sample complexity, $O((K/\varepsilon^2) \log(1/\delta))$, matches the lower bound that we derive. This learning approach substantially reduces the number of computing efficient social decisions from $K^N$ to $O(K N \log N)$, enabling us to numerically find mechanisms for $\sim 100$ players, depending on $K$.

Our contributions thus revolve around the optimization problem whose solution gives the mechanism that satisfies DE, SBB, DSIC, and IR (see Section 3-4). First, we establish a sufficient condition that ensures the optimization problem has feasible solutions, and prove that this sufficient condition is also necessary when the players have independent types (see Section 5). Second, for cases where this sufficient condition holds, we analytically derive a class of optimal solutions to this optimization problem, which in turn gives mechanisms that satisfy DE, SBB, DSIC, and IR for general environments including auctions and trading networks (see Section 5). Third, we model the problem of evaluating a quantity in the above analytical expressions as best mean estimation in MAB, propose a PAC algorithm for this problem, and prove its asymptotic optimality (see Section 6). Finally, we empirically validate the effectiveness of the proposed approach (see Section 7). In Section 2, we start by positioning our contributions to the prior work.

## 2 RELATED WORK

The prior work most related to ours is Osogami et al. (2023), which formulates and numerically solves the LP whose solution gives the mechanism for a trading network that satisfies DE, DSIC, IR, and weak budget balance (WBB; expected revenue of the mediator is nonnegative). While the objective of the LP is rather arbitrary and SBB is given just as an example in Osogami et al. (2023), we focus on SBB and drive analytical solutions for this particular objective. Our formulation extends Osogami et al. (2023) with additional parameters and notations that cover environments beyond trading networks, but these extensions are relatively straightforward.

In the rest of this section, we discuss related work on mechanism design, with a focus on those aiming to achieve SBB, as well as related work on MAB with a focus on PAC algorithms. In particular, we will see that our learning approach is unique in that we estimate a particular quantity in the optimal solution that we derive *analytically*, and this leads us to propose a new PAC algorithm and establish its optimality for an underexplored objective of best mean estimation in MAB.

In single-sided auctions where only buyers make strategic decisions, Vickrey–Clarke–Groves (VCG) mechanisms with Clark's pivot rule (also called VCG auction) satisfy *ex post* DE (called allocative efficiency in auctions), DSIC, IR, and WBB (Nisan, 2007). However, the Green-Laffont Impossibility Theorem implies that no mechanism can guarantee DE, DSIC, and SBB simultaneously for all environments (Green & Laffont, 1977; 1979). This has led to a series of work on redistributing the revenue of the mediator to the players as much as possible (i.e., to make budget balance as strong as possible), while satisfying DSIC, DE, IR, and WBB. For auctions with single or homogeneous goods (Bailey, 1997; Cavallo, 2006; Guo, 2011; Guo & Conitzer, 2007; 2009; Moulin, 2009) or for auctions where players have unit demand (Gujar & Narahari, 2011; Guo, 2012), researchers have derived analytical solutions that optimally redistribute the payment to the players. For auctions with multi-unit demands on heterogeneous goods, the prior work has proposed numerical approaches that seek to find the piecewise linear functions (Dufton et al., 2021) or neural networks (Manisha et al., 2018; Tacchetti et al., 2022) that best approximate the optimal redistribution functions.

We consider the environments that not only allow heterogeneous goods and multi-unit demands but also are more general than single-sided auctions. In particular, our players may have negative valuation on a social decision. The Myerson-Satterthwaite Impossibility Theorem (Myerson & Satterthwaite, 1983) thus implies that, unlike VCG auctions, no mechanism can guarantee *ex post* DE, DSIC, IR, and WBB simultaneously for all the environments that we consider. We thus derive mechanisms that achieve DE, DSIC, IR, and SBB in the best possible manner. A limitation of our results is that IR and SBB are satisfied only in expectation. Such a guarantee in expectation can however be justified for risk-neutral mediator and players (Osogami et al., 2023). Our model can also guarantee *strictly* positive expected utility, which in turn can ensure nonnegative utility with high probability when the player repeatedly participate in the mechanism.

For auctions, there also exists a large body of the literature on maximizing the revenue of the mediator (Myerson, 1981) with recent focus on automated mechanism design (AMD) with machine learning (Duetting et al., 2019; Rahme et al., 2021; Ivanov et al., 2022; Curry et al., 2020) and analysis of its sample complexity (Balcan et al., 2016; Morgenstern & Roughgarden, 2015; Syrgkanis, 2017). Similar to these and other studies of AMD (Sandholm, 2003; Conitzer & Sandholm, 2002), we formulate an optimization problem whose solution gives the mechanism with desired properties. However, instead of solving it numerically, we analytically derive optimal solutions. Also, while the prior work analyzes the sample complexity for finding the mechanism that maximizes the expected revenue, we analytically find optimal mechanisms and characterize the sample complexity of evaluating a particular expression in the analytically designed mechanisms.

We evaluate our analytical expression through best mean estimation (BME) in MAB, where the standard objectives are regret minimization (Auer et al., 1995; 2002) and best arm identification (BAI) (Audibert et al., 2010; Maron & Moore, 1993; Mnih et al., 2008; Bubeck et al., 2009). The prior work on MAB that is most relevant to ours is PAC learning for BAI and analysis of its sample complexity. We reduce the problem of BME to BAI and prove the lower bound on the sample complexity of BME using a technique known for BAI (Even-Dar et al., 2002). However, while this technique does not give tight lower bound for BAI (Mannor & Tsitsiklis, 2004), we show that it gives tight lower bound for BME. Notice that the problem of estimating the best mean frequently appears in reinforcement learning (van Hasselt, 2010) and machine learning (Kajino et al., 2023),

but there the focus is on how best to estimate the best mean with a given set of samples (van Hasselt, 2013), while our focus is on how best to collect samples to estimate the best mean.

## 3 SETTINGS

The goal of mechanism design is to specify the rules of a game in a way that an outcome desired by the mechanism designer is achieved when rational players (i.e., players whose goal it is to maximize their individual utility) participate in that game (Jackson, 2014; Shoham & Leyton-Brown, 2009). Formally, let $\mathcal{N} := [1, N]$ be the set of players and $\mathcal{O}$ be the set of possible outcomes. For each player $n \in \mathcal{N}$, let $\mathcal{A}_n$ be the set of available actions and $\mathcal{T}_n$ be the set of possible types. Let $\mathcal{A} := \mathcal{A}_1 \times \ldots \times \mathcal{A}_N$ and $\mathcal{T} := \mathcal{T}_1 \times \ldots \times \mathcal{T}_N$ be the corresponding product spaces. A mechanism $\mu : \mathcal{A} \to \mathcal{O}$ determines an outcome depending of the actions taken by the players. Let $u_n : \mathcal{O} \times \mathcal{T}_n \to \mathbb{R}$ be the utility function of each player $n \in \mathcal{N}$.

We consider Bayesian games where the players' types follow a probability distribution that is known to all players and the mediator. Before selecting actions, the players know their own types but not the types of the other players. A strategy of each player $n \in \mathcal{N}$ is thus a function from $\mathcal{T}_n$ to $\mathcal{A}_n$.

We assume that an outcome is determined by a social decision and payment; hence, a mechanism $\mu$ consists of a decision rule and a payment rule. Let $\mathcal{D}$ be the set of possible social decisions. Given the actions of the players, the decision rule $\phi : \mathcal{A} \to \mathcal{D}$ determines a social decision, and the payment rule $\tau : \mathcal{A} \to \mathbb{R}^{\mathcal{N}}$ determines the amount of (possibly negative) payment to the mediator from each player. Let $v : \mathcal{D} \times (\mathcal{T}_1 \cup \ldots \cup \mathcal{T}_N) \to \mathbb{R}$ specify the value of a given social decision to the player of a given type. Then the utility of player $i$ when players take actions $a \in \mathcal{A}$ is

$$u_n(\mu(a); t_n) = u_n((\phi(a), \tau(a)); t_n) = v(\phi(a); t_n) - \tau_n(a). \tag{1}$$

Throughout, we assume that $\mathcal{N}, \mathcal{D}$, and $\mathcal{T}_n, \forall n \in \mathcal{N}$ are finite sets.

Without loss of generality by the revelation principle (Shoham & Leyton-Brown, 2009), we consider only direct mechanisms, where the action available to each player is to declare which type the player belongs to from the set of possible types (i.e., $\mathcal{A}_n = \mathcal{T}_n, \forall n \in \mathcal{N}$). We will thus use $\mathcal{T}_n$ for $\mathcal{A}_n$.

Then we seek to achieve the following four properties with our mechanisms. The first property is Dominant Strategy Incentive Compatibility (DSIC), which ensures that the optimal strategy of each player is to truthfully reveal its type regardless of the strategies of the other players. Formally,

$$[\text{DSIC}] \; v(\phi(t_n, t'_{-n}); t_n) - \tau_n(t_n, t'_{-n}) \geq v(\phi(t'); t_n) - \tau_n(t'), \forall t' \in \mathcal{T}, \forall t_n \in \mathcal{T}_n, \forall n \in \mathcal{N}, \tag{2}$$

where the left-hand side represents the utility of the player having type $t_n$ when it declares the same $t_n$, and the other players declare arbitrary types $t'_{-n}$.

The second property is Decision Efficiency (DE), which requires that the mediator chooses the social decision that maximizes the total value to the players. With DSIC, we can assume that the players declare true types, and hence we can write DE as a condition on the decision rule:

$$[\text{DE}] \qquad \phi(t) \in \operatorname*{argmax}_{d \in \mathcal{D}} \sum_{n \in \mathcal{N}} v(d; t_n) \qquad \forall t \in \mathcal{T}. \tag{3}$$

As the third property, we generalize individual rationality and require that the expected utility of each player is at least as large as a target value that can depend on its type. We refer to this property as $\theta$-IR. Again, assuming that players declare true types due to DSIC, we can write $\theta$-IR as follows:

$$[\theta\text{-IR}] \qquad \mathbb{E}[v(\phi(t); t_n) - \tau_n(t) \mid t_n] \geq \theta(t_n) \qquad \forall t_n \in \mathcal{T}_n, \forall n \in \mathcal{N}, \tag{4}$$

where $\theta : \mathcal{T}_1 \cup \ldots \cup \mathcal{T}_N \to \mathbb{R}$ determines the target expected utility for each type. Throughout (except in Section 6, where we discuss general MAB models), $\mathbb{E}$ denotes the expectation with respect to the probability distribution $\mathbb{P}$ of types, which is the only probability that appears in our mechanisms.

The last property is a generalization of Budget Balance (BB), which we refer to as $\rho$-WBB and $\rho$-SBB. Specifically, $\rho$-WBB requires that the expected revenue of the mediator is no less than a given constant $\rho \in \mathbb{R}$, and $\rho$-SBB requires that it is equal to $\rho$. Again, assuming that the players declare true types due to DSIC, these properties can be written as follows:

$$[\rho\text{-WBB}] \qquad \sum_{n \in \mathcal{N}} \mathbb{E}\left[\tau_n(t)\right] \geq \rho. \qquad\qquad [\rho\text{-SBB}] \qquad \sum_{n \in \mathcal{N}} \mathbb{E}\left[\tau_n(t)\right] = \rho. \tag{5}$$

While $\rho$-SBB is stronger than $\rho$-WBB, we will see that $\rho$-SBB is satisfiable iff $\rho$-WBB is satisfiable.

# 4 OPTIMIZATION PROBLEM FOR AUTOMATED MECHANISM DESIGN

Following Osogami et al. (2023), we seek to find optimal mechanisms in the class of VCG mechanisms. A VCG mechanism is specified by a pair $(\phi^\star, h)$. Specifically, after letting player take the actions of declaring their types $t \in \mathcal{T}$, the mechanism first finds a social decision $\phi^\star(t)$ using a decision rule $\phi = \phi^\star$ that satisfies DE (3). It then determines the amount of payment from each player $n \in \mathcal{N}$ to the mediator by

$$\tau_n(t) = h_n(t_{-n}) - \sum_{m \in \mathcal{N}_{-n}} v(\phi^\star(t); t_m), \tag{6}$$

where we define $\mathcal{N}_{-n} := \mathcal{N} \setminus \{n\}$, and $h_n : \mathcal{T}_{-n} \to \mathbb{R}$ is an arbitrary function of the types of the players other than $n$ and referred to as a pivot rule. The decision rule $\phi^\star$ guarantees DE (3) by construction, and the payment rule (6) then guarantees DSIC (2) (Nisan, 2007).

Our problem is now reduced to find the pivot rule, $h = \{h_n\}_{n \in \mathcal{N}}$, that minimizes the expected revenue of the mediator, while satisfying $\theta$-IR and $\rho$-WBB. This may lead to satisfying $\rho$-SBB if the revenue is maximally reduced. To represent this reduced problem, let

$$w^\star(t) := \sum_{n \in \mathcal{N}} v(\phi^\star(t); t_n) \tag{7}$$

be the total value of the efficient social decision when the players have types $t$. Then we can rewrite $\theta$-IR (for the player having type $t_n$) and $\rho$-WBB as follows (see Appendix A.1 for full derivation):

$$\mathbb{E}[v(\phi^\star(t); t_n) - \tau_n(t) \mid t_n] \geq \theta(t_n) \iff \mathbb{E}[w^\star(t) \mid t_n] - \mathbb{E}[h_n(t_{-n}) \mid t_n] \geq \theta(t_n) \tag{8}$$

$$\sum_{n \in \mathcal{N}} \mathbb{E}[\tau_n(t)] \geq \rho \iff \sum_{n \in \mathcal{N}} \mathbb{E}[h_n(t_{-n})] - (N-1)\mathbb{E}[w^\star(t)] \geq \rho. \tag{9}$$

Therefore, we arrive at the following linear program (LP):

$$\min_h \quad \sum_{n \in \mathcal{N}} \mathbb{E}[h_n(t_{-n})] \tag{10}$$

$$\text{s.t.} \quad \mathbb{E}[w^\star(t) \mid t_n] - \mathbb{E}[h_n(t_{-n}) \mid t_n] \geq \theta(t_n) \qquad \forall t_n \in \mathcal{T}_n, \forall n \in \mathcal{N} \tag{11}$$

$$\sum_{n \in \mathcal{N}} \mathbb{E}[h_n(t_{-n})] - (N-1)\mathbb{E}[w^\star(t)] \geq \rho. \tag{12}$$

The approach of Osogami et al. (2023) is to numerically solve this LP possibly with approximations. Since $h_n(t_{-n})$ is a variable for each $t_{-n} \in \mathcal{T}_{-n}$ and each $n \in \mathcal{N}$, the LP has $N K^{N-1}$ variables and $N K + 1$ constraints, when each player has $K$ possible types. When this LP is feasible, let $h^\star$ be its optimal solution; then the VCG mechanism $(\phi^\star, h^\star)$ guarantees DSIC, DE, $\theta$-IR, and $\rho$-WBB (formalized as Proposition 4 in Appendix A.1). Otherwise, no VCG mechanisms can guarantee them all. In the next section, we characterize exactly when the LP is feasible and provide analytical solutions to the LP.

# 5 ANALYTICAL SOLUTION TO THE OPTIMIZATION PROBLEM

We first establish a sufficient condition and a necessary condition for the LP to have feasible solutions. Note that complete proofs for all theoretical statements are provided in Appendix B.

**Lemma 1.** *The LP* (10)-(12) *is feasible if*

$$\sum_{n \in \mathcal{N}} \min_{t_n \in \mathcal{T}_n} \{\mathbb{E}[w^\star(t) \mid t_n] - \theta(t_n)\} \geq (N-1)\mathbb{E}[w^\star(t)] + \rho. \tag{13}$$

**Lemma 2.** *When types are independent* ($t_m$ *and* $t_n$ *are independent for any* $m \neq n$ *under* $\mathbb{P}$*), the LP* (10)-(12) *is feasible only if* (13) *holds.*

These two lemmas establish the following necessary and sufficient condition.

**Corollary 1.** *When types are independent, the LP* (10)-(12) *is feasible if and only if* (13) *holds.*

While the LP is not necessarily feasible, one may choose $\theta$ and $\rho$ in a way that it is guaranteed to be feasible. For example, the feasibility is guaranteed (i.e., (13) is satisfied) by setting

$$\theta \equiv 0 \qquad \text{and} \qquad \rho = \left[ \sum_{n \in \mathcal{N}} \min_{t_n \in \mathcal{T}_n} \mathbb{E}[w^{\star}(t) \mid t_n] - (N-1) \, \mathbb{E}[w^{\star}(t)] \right]^{-}, \qquad (14)$$

where $[x]^{-} := \min\{x, 0\}$ for $x \in \mathbb{R}$. When $\rho < 0$, the mediator might get negative expected revenue, but the expected loss of the mediator is at most $|\rho|$. Appendix A.2 provides alternative $\rho$ and $\theta$ that guarantee feasibility, but some of the players might incur negative expected utility.

Finally, we derive a class of optimal solutions to the LP when it is feasible.

**Lemma 3.** *A pivot rules is said to be constant if and only if, for each $n \in \mathcal{N}$, there exists a constant $\eta_n$ such that $h_n(t_{-n}) = \eta_n, \forall t_{-n} \in \mathcal{T}_{-n}$. Let $\mathcal{H}$ be the set of constant pivot rules that satisfy:*

$$\eta_n = \min_{t_n \in \mathcal{T}_n} \{\mathbb{E}[w^{\star}(t) \mid t_n] - \theta(t_n)\} - \delta_n \qquad \forall n \in \mathcal{N} \qquad (15)$$

*where $\delta$ lies on the following simplex:*

$$\delta_n \geq 0 \qquad \forall n \in \mathcal{N} \qquad (16)$$

$$\sum_{n \in \mathcal{N}} \delta_n = \sum_{n \in \mathcal{N}} \min_{t_n \in \mathcal{T}_n} \{\mathbb{E}[w^{\star}(t) \mid t_n] - \theta(t_n)\} - (N-1) \, \mathbb{E}[w^{\star}(t)] - \rho. \qquad (17)$$

*Then, when (13) holds, $\mathcal{H}$ is nonempty, and any $h \in \mathcal{H}$ is an optimal solution to the LP (10)-(12).*

When types are independent, the condition (13) is necessary for the existence of a feasible solution; hence, we do not lose optimality by considering only the solutions in $\mathcal{H}$. On the other hand, when types are dependent, the condition (13) may still be satisfied, where the solutions in $\mathcal{H}$ remain optimal. However, dependent types no longer necessitate (13) to satisfy (11)-(12); hence the space of constant pivot rules may not suffice to find an optimal solution if (13) does not hold.

**Proposition 1.** *The LP (10)-(12) may be feasible even if (13) is violated (when types are dependent).*

**Corollary 2.** *Feasible solutions to the LP (10)-(12) include at least one constant pivot rule (defined in Lemma 3) if and only if (13) holds (whether types are dependent or not).*

In deriving the optimal solutions, we have substantially reduced the essential number of variables (from $N \, T^{N-1}$ to $N$ when each player has $T$ types). Our approach can therefore not only find but also represent or store solutions exponentially more efficiently than Osogami et al. (2023).

It turns out that the solutions in $\mathcal{H}$ not only satisfy $\rho$-WBB but also $\rho$-SBB (in addition to DE, DSIC, and $\theta$-IR) regardless of whether the types are independent or not:

**Corollary 3.** *Any VCG mechanism given with a pivot rule in $\mathcal{H}$ satisfies $\rho$-SBB.*

When the LP is infeasible, we can construct a mechanism that satisfies one of $\rho$-SBB and $\theta$-IR (in addition to DE and DSIC) regardless of whether the types are independent or not:

**Corollary 4.** *Any VCG mechanism with a pivot rule given by (15) and (17) satisfies $\rho$-SBB.*

**Corollary 5.** *Any VCG mechanism with a pivot rule given by (15) and (16) satisfies $\theta$-IR for any $t_n \in \mathcal{T}_n$ and $n \in \mathcal{N}$.*

For example, for any $t_n \in \mathcal{T}_n$ and $n \in \mathcal{N}$, the following pivot rule always satisfies $\theta$-IR:

$$\eta_n = \min_{t_n \in \mathcal{T}_n} \{\mathbb{E}[w^{\star}(t) \mid t_n] - \theta(t_n)\} - \max\{\delta, 0\}, \qquad (18)$$

where we define (see also (58) in the appendix)

$$\delta := \frac{1}{N} \left( \sum_{n \in \mathcal{N}} \min_{t_n \in \mathcal{T}_n} \{\mathbb{E}[w^{\star}(t) \mid t_n] - \theta(t_n)\} - (N-1) \, \mathbb{E}[w^{\star}(t)] - \rho \right). \qquad (19)$$

So far, we have analytically derived the class of optimal solutions for the LP. Although these analytical solutions substantially reduce the computational cost of solving the LP compared to the numerical solutions in Osogami et al. (2023), they still need to evaluate

$$\min_{t_n \in \mathcal{T}_n} \{\mathbb{E}[w^{\star}(t) \mid t_n] - \theta(t_n)\} \qquad (20)$$

for each $n \in \mathcal{N}$. Since $\mathbb{E}$ is the expectation with respect to the distribution $\mathbb{P}$ over $\mathcal{T}$, this would require evaluating $w^{\star}(t)$ for all $t \in \mathcal{T}$. Recall that $w^{\star}(t)$ defined with (7) is the total value of the efficient decision $\phi^{\star}(t)$, which is given by a solution to an optimization problem (3). Without any structure in $\mathcal{D}$ and $v$, this would require evaluating the total value for all decisions in $\mathcal{D}$.

## 6  ONLINE LEARNING FOR EVALUATING THE ANALYTICAL SOLUTION

To alleviate the computational complexity associated with (20), we propose a learning approach. A key observation is that the problem of estimating (20) can be considered as a variant of a multi-armed bandit (MAB) problem whose objective is to estimate the mean reward of the best arm. Specifically, the MAB gives the reward $\theta(t_n) - w^{\star}(t)$ when we pull an arm $t_n \in \mathcal{T}_n$, where $t$ is a sample from the conditional distribution $\mathbb{P}[\cdot \mid t_n]$. Following the relevant prior work on MAB (Even-Dar et al., 2002; 2006; Hassidim et al., 2020; Mannor & Tsitsiklis, 2004), we maximize reward in this section.

Since we assume that $\mathcal{N}$, $\mathcal{D}$, and $\mathcal{T}_n, \forall n \in \mathcal{N}$ are finite, there exist constants, $\bar{\theta}$ and $\bar{v}$, such that $|\theta(t')| \leq \bar{\theta}$ and $|v(d; t')| \leq \bar{v}, \forall d \in \mathcal{D}, \forall t' \in \cup_{n \in \mathcal{N}} \mathcal{T}_n$. Then we can also bound $|\theta(t_n) - w^{\star}(t)| \leq \bar{\theta} + N \bar{v}, \forall t \in \mathcal{T}, \forall n \in \mathcal{N}$. Namely, the reward in the MAB is bounded. We assume that we know the bounds on the reward and can scale it such that the scaled reward is in $[0, 1]$ surely.

We also assume that we have access to an arbitrary size of the sample that is independent and identically distributed (i.i.d.) according to $\mathbb{P}[\cdot \mid t_n]$ for any $t_n \in \mathcal{T}_n, n \in \mathcal{N}$. When players have independent types, such sample can be easily constructed as long as we have access to i.i.d. sample $\{t^{(i)}\}_{i=1,2,\ldots}$ from $\mathcal{T}$, because $\{(t_n, t_{-n}^{(i)})\}_i$ is the sample from $\mathbb{P}[\cdot \mid t_n]$ for any $t_n \in \mathcal{T}_n, n \in \mathcal{N}$.

Consider the general $K$-armed bandit where the reward of each arm is bounded in $[0, 1]$. For each $k \in [1, K]$, let $\mu_k$ be the true mean of arm $k$. Let $\mu_{\star} := \max_k \mu_k$ be the best mean-reward, which we want to estimate. We say that the sample complexity of an algorithm for a MAB is $T$ if the sample size needed by the algorithm is bounded by $T$ (i.e., arms are pulled at most $T$ times).

A standard Probably Approximately Correct (PAC) algorithm for MAB returns an $\varepsilon$-optimal arm with probability at least $1 - \delta$ for given $\varepsilon, \delta$ (Even-Dar et al., 2006; Hassidim et al., 2020). We will use the following definition:

**Definition 1** (($\varepsilon, \delta$)-PAC Best Arm Identifier (Even-Dar et al., 2006))**.** *For $\varepsilon, \delta > 0$, we say that an algorithm is ($\varepsilon, \delta$)-PAC Best Arm Identifier (BAI) if the arm $\hat{I}$ identified by the algorithm satisfies*

$$\Pr\left(\mu_{\hat{I}} \geq \mu_{\star} - \varepsilon\right) \geq 1 - \delta. \tag{21}$$

Definition 1 is different from what we need to evaluate (20). Formally, we need

**Definition 2** (($\varepsilon, \delta$)-PAC Best Mean Estimator)**.** *For $\varepsilon, \delta > 0$, we say that an algorithm is ($\varepsilon, \delta$)-PAC Best Mean Estimator (BME) if the best mean $\hat{\mu}$ estimated by the algorithm satisfies*

$$\Pr\left(|\hat{\mu} - \mu_{\star}| \leq \varepsilon\right) \geq 1 - \delta. \tag{22}$$

BAI and BME are related but different. For example, consider the case where the best arm has large variance and $\mu_{\star} = 1/2$, and the other arms always give zero reward $\mu_n = 0, \forall n \neq \star$. Then relatively small sample would suffice for BAI due to the large gap $\mu_{\star} - \mu_n = 1/2, \forall n \neq \star$, while BME would require relatively large sample due to the large variance of the best arm $\star$. As another example, consider the case where we have many arms whose rewards follow Bernoulli distributions. Suppose that half of the arms have an expected value of 1, and the other half have an expected value of $1 - (3/2)\varepsilon$. Then by pulling sufficiently many arms (once for each arm), we can estimate that the best mean is at least $1 - \varepsilon$ with high probability (by Hoeffding's inequality), but BAI would require pulling arms sufficiently many times until we pull one of the best arms sufficiently many times (to be able to say that this particular arm has an expected value at least $1 - \varepsilon$).

For ($\varepsilon, \delta$)-PAC BAI, the following lower and upper bounds on the sample complexity are known:

**Proposition 2** (Mannor & Tsitsiklis (2004))**.** *There exists a ($\varepsilon, \delta$)-PAC BAI with sample complexity $O((K/\varepsilon^2) \log(1/\delta))$, and any ($\varepsilon, \delta$)-PAC BAI must have the sample complexity at least $\Omega((K/\varepsilon^2) \log(1/\delta))$.*

We establish the same lower and upper bounds for $(\varepsilon, \delta)$-PAC BME:

**Theorem 1.** *There exists a $(\varepsilon, \delta)$-PAC BME with sample complexity $O((K/\varepsilon^2) \log(1/\delta))$, and any $(\varepsilon, \delta)$-PAC BME must have the sample complexity at least $\Omega((K/\varepsilon^2) \log(1/\delta))$.*

Our upper bound is established by reducing BME to BAI. Suppose that we have access to an arbitrary $(\varepsilon, \delta)$-PAC BAI with sample complexity $M$ (Algorithm 2 in Appendix A.3). We can construct a $(\frac{3}{2}\varepsilon, 2\delta)$-PAC BME by first running the $(\varepsilon, \delta)$-PAC BAI and then sampling from the arm $\hat{I}$ that is identified as the best until we have a sufficient number, $m^\star$, of samples from $\hat{I}$ (Algorithm 3 in Appendix A.3). When $m^\star$ is appropriately selected, the following lemma holds:

**Lemma 4.** *When a $(\varepsilon, \delta)$-PAC BAI with sample complexity $M$ is used, Algorithm 3 has sample complexity $M + m^\star$, where $m^\star := \lceil (2/\varepsilon^2) \log(1.22/\delta) \rceil$, and returns $\hat{\mu}_{\hat{I}}$ that satisfies*

$$\Pr\left(|\hat{\mu}_{\hat{I}} - \mu_\star| > \frac{3}{2}\varepsilon\right) \le 2\,\delta. \tag{23}$$

By Proposition 2, this establishes the upper bound on the sample complexity in Theorem 1.

Our lower bound is established by showing that an arbitrary $(\varepsilon, \delta)$-PAC BME can be used to solve the problem of identifying whether a given coin is negatively or positively biased (precisely, $\varepsilon$-Biased Coin Problem of Definition 3 in Appendix A.3), for which any algorithm is known to require *expected* sample complexity at least $\Omega((1/\varepsilon^2) \log(1/\delta))$ to solve correctly with probability at least $1 - \delta$ (Chernoff, 1972; Even-Dar et al., 2002) (see Lemma 7 in Appendix A.3). The following lemma, together with Lemma 7, establishes the lower bound on the sample complexity in Theorem 1:

**Lemma 5.** *If there exists an $(\varepsilon/2, \delta/2)$-PAC BME with sample complexity $M$ for $K$-armed bandit, then there also exists an algorithm, having expected sample complexity $M/K$, that can solve the $\varepsilon$-Biased Coin Problem correctly with probability at least $1 - \delta$.*

We prove Lemma 5 by reducing the $\varepsilon$-Biased Coin Problem to BME. This proof technique was also used in Even-Dar et al. (2002) to prove a lower bound on the sample complexity of BAI. What is interesting, however, is that the lower bound in Even-Dar et al. (2002) is not tight when $\delta < 1/K$, and a tight lower bound is established by a different technique in Mannor & Tsitsiklis (2004). On the other hand, our proof gives a tight lower bound on the sample complexity of BME. In Appendix A.3, we further discuss where this difference in the derived lower bounds comes from.

Finally, we connect the results on BME back to our mechanism in Section 5 and provide a guarantee on the properties of the mechanism when the term (20) is estimated with BME. To this end, we define the terms involving expectations in Lemma 3 as follows:

$$\kappa_n(\theta) := \min_{t_n \in \mathcal{T}_N} \{\mathbb{E}[w^\star \mid t_n] - \theta(t_n)\} \tag{24}$$

$$\lambda(\rho) := \mathbb{E}[w^\star(t)] + \rho/(N-1). \tag{25}$$

Then, recalling that $N$ is the number of players, we have the following lemma:

**Lemma 6.** *Let $\tilde{\kappa}_n(\theta)$ for $n \in \mathcal{N}$ and $\tilde{\lambda}(\rho)$ be independent estimates of $\kappa_n(\theta)$ and $\lambda(\rho)$ respectively given by an $(\varepsilon', \delta')$-PAC Best Mean Estimator and a standard $(\varepsilon'', \delta')$-PAC estimator of expectation. Also, let $\tilde{d} := d(\tilde{\kappa}(\theta), \tilde{\lambda}(\rho), \varepsilon''', \varepsilon'''')$ be a point on the following simplex (here, we change the notation from $\delta$ in Lemma 3 to $\tilde{d}$ to avoid confusion):*

$$\tilde{d}_n \ge \varepsilon''', \forall n \in \mathcal{N} \tag{26}$$

$$\sum_{n \in \mathcal{N}} \tilde{d}_n = \sum_{n \in \mathcal{N}} \tilde{\kappa}_n(\theta) - (N-1)(\tilde{\lambda}(\theta) + \varepsilon'''') \tag{27}$$

*Then the VCG mechanism with the constant pivot rule $h_n(t_{-n}) = \eta_n = \tilde{\kappa}_n(\theta) - \tilde{d}_n$ satisfies $(\theta - (\varepsilon' - \varepsilon'''))$-IR and $(\rho - (N-1)(\varepsilon'' - \varepsilon''''))$-WBB with probability $(1 - \delta')^{N+1}$.*

Notice that the sufficient condition of Lemma 1 states that the simplex in Lemma 3 is nonempty. Analogously, when the simplex in Lemma 6 is empty, we cannot provide the solution that guarantees the properties stated in Lemma 6. Since DSIC and DE remain satisfied regardless of whether $\kappa_n$ for $n \in \mathcal{N}$ and $\lambda$ are estimated or exactly computed, Lemma 6 immediately establishes the following theorem by appropriately choosing the parameters:

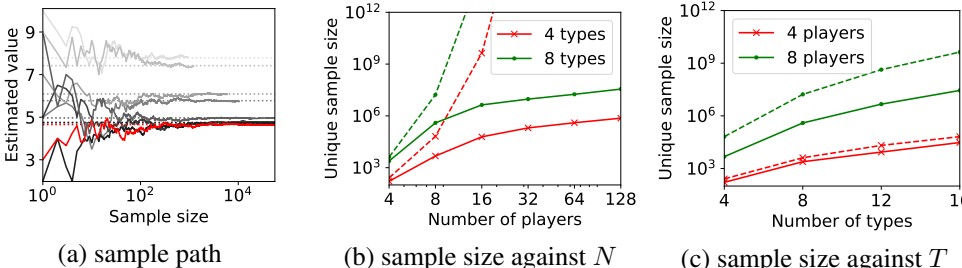

(a) sample path      (b) sample size against $N$      (c) sample size against $T$

Figure 1: (a) Representative sample path that shows the estimated values of $\mathbb{E}[w^\star(t) \mid t_n]$ for $t_n \in \mathcal{T}_n$ against sample size used by a $(0.25, 0.1)$-PAC BME with Successive Elimination (SE-BME; Algorithm 4) for 8 players, each with 8 types. (b)-(c) The unique sample size (the number of unique $t$ which $w^\star(t)$ is evaluated with) required by the exact computation of $\min_{t_n \in \mathcal{T}_n} \mathbb{E}[w^\star(t) \mid t_n], \forall n \in \mathcal{N}$ (dashed curves) and by SE-BME (solid curves).

**Theorem 2.** *In Lemma 6, let $\varepsilon''' = \varepsilon'$, $\varepsilon'''' = \varepsilon''$, and $\delta' = 1 - (1 - \delta)^{1/(N+1)}$. Then the VCG mechanism with the constant pivot rule $h_n(t_{-n}) = \tilde{\kappa}_n(\theta) - \tilde{d}_n$ satisfies DSIC, DE, $\theta$-IR, and $\rho$-WBB with probability $1 - \delta$.*

Recall that the exact computation of our mechanism in Lemma 3 requires evaluating $w^\star(t)$ for all $t \in \mathcal{T}$, whose size grows exponentially with the number of players, $N$. Our BME algorithm reduces this requirement to $O(N \log N)$, as is formally proved in the following proposition:

**Proposition 3.** *The sample complexity to learn the constant pivot rule in Theorem 2 is $O((N K / \varepsilon^2) \log(N/\delta))$, where $N = |\mathcal{N}|$ is the number of players, and $K = \max_{n \in \mathcal{N}} \mathcal{T}_n$ is the maximum number of types of each player.*

## 7 NUMERICAL EXPERIMENTS

We conduct several numerical experiments to validate the effectiveness of the proposed approach and to understand its limitations. Specifically, we design our experiments to investigate the following questions. i) BME studied in Section 6 has asymptotically optimal sample complexity, but how well can we estimate the best mean when there are only a moderate number of arms? ii) How much can BME reduce the number of times $w^\star(t)$ given by (7) is evaluated? iii) How well $\theta$-IR and $\rho$-SBB are satisfied when (20) is estimated by BME rather than calculated exactly? In this section, we only provide a brief overview of our experiments; for full details, see Appendix A.4. In particular, we find the key empirical property of BME that it can reduce the computational complexity by many orders of magnitude when the number of players is moderate ($\sim 10$) to large but is relatively less effective against the increased number of types.

For question i), we compare Successive Elimination for BAI (SE-BAI), which is known to perform well for a moderate number of arms (Hassidim et al., 2020; Even-Dar et al., 2006), against the corresponding algorithm for BME (SE-BME) and summarize the results in Appendix A.4.1. Overall, we find that SE-BME generally requires smaller sample size than SE-BAI, except when there are only a few arms and their means have large gaps. The efficiency of SE-BAI for this case makes intuitive sense, since the best arm can be identified without estimating the means with high accuracy.

For question ii), we quantitatively validate the effectiveness of SE-BME in reducing the number of times $w^\star(t)$ is computed when we evaluate (20) in a setting of mechanism design, as is detailed in Appendix A.4.2. Figure 1 summarizes the results. Panel (a) shows that the means close to the minimum value are estimated with high accuracy, while others are eliminated by SE-BME after a small number of samples. Panels (b)-(c) show that SE-BME (solid curves) evaluates $w^\star(t)$ by orders of magnitude smaller number of times than what is required by exact computation (dashed curves). The effectiveness of SE-BME is particularly striking when $N$ (the number of players) grows. While exact computation evaluates $w^\star(t)$ exponentially many times as $N$ grows, SE-BME has only linear dependency on $N$. This insensitivity of the sample complexity of SE-BME to $N$ makes intuitive sense, since $N$ only affects the distribution of the reward and keeps the number of arms unchanged.

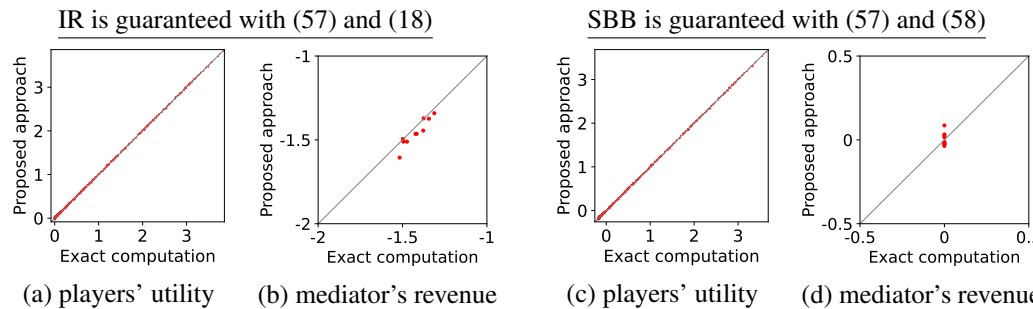

Figure 2: The red dot shows expected utility in (a) and (c) and expected revenue in (b) and (d), when analytical solutions are evaluated exactly (horizontal) or estimated with $(0.25, 0.1)$-PAC SE-BME (vertical) for environments with 8 players, each having 8 possible types. The analytical solution guarantees IR in (a) and (b) and SBB in (c) and (d). Results are plotted for 10 random seeds.

For question iii), we quantitatively evaluate how well $\theta$-IR and $\rho$-SBB are guaranteed when we estimate (20) with SE-BME, where we continue to use the same setting of mechanism design. In Figure 2, we study two analytical solutions: one guarantees to satisfy $\theta$-IR (with $\theta \equiv 0$), and the other guarantees to satisfy $\rho$-SBB (with $\rho = 0$). The red dots show the expected utility of the players in (a) and (c) as well as the expected revenue of the mediator in (b) and (d), when (20) is evaluated either exactly (horizontal axes) or estimated with SE-BME (vertical axes). Overall, the expected revenue of the mediator is more sensitive than the expected utility of the players to the error in the estimation of (20), because (12) involves the summation $\sum_{n \in \mathcal{N}} \eta_n$, while (11) only involves $\eta_n$ for a single $n \in \mathcal{N}$. We can however ensure that $\rho$-WBB (instead of SBB) is satisfied with high probability by replacing the $\rho$ with a $\rho' > \rho$ when we compute $\eta_n$ after (20) is estimated. As we show and explain with Figure 8 in Appendix A.4.3, this will simply shift the dots in the figure, and we can guarantee $\rho$-WBB with high probability with an appropriate selection of $\rho'$.

We have run all of the experiments on a single core with at most 66 GB memory without GPUs in a cloud environment, as is detailed in Appendix A.4.4. The associated source code is submitted as a supplementary material and will be open-sourced upon acceptance.

# 8 CONCLUSION

We have analytically derived optimal solutions for the LP that gives mechanisms that guarantee the desired properties of DE, DSIC, $\rho$-SBB, and $\theta$-IR. When there are $N$ players, each with $T$ possible types, the LP involves $N T^{N-1}$ variables, while our analytical solutions are represented by only $N$ essential variables. While Osogami et al. (2023) numerically solves this LP only for $N = T = 2$, we have exactly evaluated our analytical solutions for $N = T = 8$ (see Figure 1). The analytical solution, however, involves a term whose exact evaluation requires finding efficient social decisions $T^N$ times. We have modeled the problem of evaluating this term as best mean estimation in multi-armed bandit, proposed a PAC estimator, and proved its asymptotic optimality by establishing a lower bound on the sample complexity. Our experiments show that our PAC estimator enables finding mechanism for $N = 128$ and $T = 8$ with a guarantee on the desired properties.

The proposed approach makes a major advancement in the field and can positively impact society by providing guarantees on desired properties for large environments, which existing approaches are unable to handle. However, it presents certain limitations and potential challenges, which inspire several directions for further research. Regarding limitations, one should keep in mind that our mechanisms guarantee $\rho$-SBB and $\theta$-IR only in expectation, and that the our approaches have only been applied to the environments up to 128 players and up to 16 types in our experiments. Regarding societal impacts, our approach might result in the mechanisms that are unfair to some of the players, because DE does not mean that all of the players are treated fairly. These limitations and societal impacts are further discussed in Appendix A.5.

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

# A DETAILS

In this section, we provide full details of derivations and other details skipped in the body of the paper.

## A.1 DETAILS OF SECTION 4

Equivalence in (8) follows from

$$\mathbb{E}[v(\phi^\star(t); t_n) - \tau_n(t) \mid t_n] \geq \theta(t_n)$$

$$\iff \mathbb{E}\left[\sum_{m \in \mathcal{N}} v(\phi^\star(t); t_m) \middle| t_n\right] - \mathbb{E}[h_n(t_{-n}) \mid t_n] \geq \theta(t_n) \tag{28}$$

$$\iff \mathbb{E}[w^\star(t) \mid t_n] - \mathbb{E}[h_n(t_{-n}) \mid t_n] \geq \theta(t_n). \tag{29}$$

Equivalence in (9) follows from

$$\sum_{n \in \mathcal{N}} \mathbb{E}[\tau_n(t)] \geq \rho \iff \sum_{n \in \mathcal{N}} \mathbb{E}[h_n(t_{-n})] - \sum_{n \in \mathcal{N}} \sum_{m \in \mathcal{N}_{-n}} \mathbb{E}[v(\phi^\star(t); t_m)] \geq \rho \tag{30}$$

$$\iff \sum_{n \in \mathcal{N}} \mathbb{E}[h_n(t_{-n})] - (N-1)\mathbb{E}[w^\star(t)] \geq \rho, \tag{31}$$

where the last equivalence follows from the definition of $w^\star(t)$ in (7).

**Proposition 4.** *Let the decision rule $\phi^\star$ be the one that satisfies (3) and the payment rule $\tau = (\tau_n)_n$ be in the form of (6) where $h = (h_n)_n = (h_n^\star)_n = h^\star$ is given by the solution to the LP (10)-(12). Then the VCG mechanism $(\phi^\star, \tau^\star)$ satisfies DSIC, DE, $\theta$-IR, and $\rho$-WBB.*

*Proof.* With the equivalences (8)-(9), the constraints (11)-(12) in the LP guarantee that $\theta$-IR and $\rho$-WBB are satisfied by feasible solutions. Since we consider the class of VCG mechanisms, DE is trivially satisfied by the definition of $\phi^\star$, and DSIC is satisfied when the payment rule is in the form of (6). Hence, all of DSIC, DE, $\theta$-IR, and $\rho$-WBB are satisfied by $(\phi^\star, \theta^\star)$. $\qquad\square$

Algorithm 1 summarizes the protocol under the VCG mechanism discussed Section 4. In Step 3, the optimal strategy of each player is to truthfully declare its type $\hat{t}_n = t_n$. In Step 5, the LP may not be feasible, in which case the protocol may fail, or we may use another payment rule to proceed.

---

**Algorithm 1** Protocol under the VCG mechanism

---

1: Sample the type profile $t$ from the common prior $\mathbb{P}$
2: Each player $n \in \mathcal{N}$ gets to know its own type $t_n$
3: Each player $n \in \mathcal{N}$ declare their type $\hat{t}_n$
4: Determine the social decision: $\phi^\star(\hat{t}) \leftarrow \underset{d \in \mathcal{D}}{\arg\max} \sum_{n \in \mathcal{N}} v(d; \hat{t}_n)$

5: $h^\star \leftarrow$ Find the optimal solution to the LP (10)-(12)
6: Determine the payment from each player $n \in \mathcal{N}$ to the mediator:
$$\tau_n(\hat{t}) \leftarrow h_n^\star(\hat{t}_{-n}) - \sum_{m \in \mathcal{N}_{-n}} v(\phi^\star(\hat{t}); \hat{t}_m)$$
7: Each player $n \in \mathcal{N}$ gets utility $v(\phi^\star(\hat{t}); t_n) - \tau_n(\hat{t})$

---

## A.2 DETAILS OF SECTION 5

### A.2.1 SOLUTIONS THAT GUARANTEE WBB

Alternatively, one may set

$$\theta(t_n) = \left[\mathbb{E}[w^\star(t) \mid t_n] - \frac{N-1}{N}\mathbb{E}[w^\star(t)]\right]^- \qquad \forall t_n \in \mathcal{T}_n, \forall n \in \mathcal{N} \tag{32}$$

$$\rho = 0 \tag{33}$$

---

**Algorithm 2** `PAC-BAI`$(\varepsilon, \delta, T)$

---

1: Run an arbitrary $(\varepsilon, \delta)$-PAC BAI with sample complexity $T$
2: Let $\hat{I}$ be the $\varepsilon$-optimal arm identified by the algorithm
3: Let $M_{\hat{I}}$ be the number of times the algorithm has pulled arm $\hat{I}$; if no such information is available, set $M_{\hat{I}} = 0$
4: Let $\hat{\mu}_{\hat{I}}$ be the sample average of the $M_{\hat{I}}$ rewards obtained from arm $\hat{I}$; if $M_{\hat{I}} = 0$, set $\hat{\mu}_{\hat{I}} = 0$
5: **return** $\hat{I}, M_{\hat{I}}, \hat{\mu}_{\hat{I}}$

---

**Algorithm 3** Best Mean Estimator

---

1: $\hat{I}, M_{\hat{I}}, \hat{\mu}_{\hat{I}} \leftarrow$ `PAC-BAI`$(\varepsilon, \delta, T)$
2: **if** $M_{\hat{I}} < m^{\star}$, where $m^{\star}$ is defined in Lemma 4 **then**
3:      Pull arm $\hat{I}$ for $m^{\star} - M_{\hat{I}}$ times
4:      Update the sample average $\hat{\mu}_{\hat{I}}$ of arm $\hat{I}$
5: **end if**
6: **return** $\hat{\mu}_{\hat{I}}$

---

to guarantee the feasibility, since

$$
\sum_{n \in \mathcal{N}} \min_{t_n \in \mathcal{T}_n} \left\{ \mathbb{E}[w \mid t_n] - \left[ \mathbb{E}[w^{\star}(t) \mid t_n] - \frac{N-1}{N} \mathbb{E}[w^{\star}(t)] \right]^{-} \right\} - (N-1) \mathbb{E}[w^{\star}(t)]
$$

$$
= \sum_{n \in \mathcal{N}} \left( \min_{t_n \in \mathcal{T}_n} \left\{ \mathbb{E}[w \mid t_n] - \frac{N-1}{N} \mathbb{E}[w^{\star}(t)] - \left[ \mathbb{E}[w^{\star}(t) \mid t_n] - \frac{N-1}{N} \mathbb{E}[w^{\star}(t)] \right]^{-} \right\} \right) \quad (34)
$$

$$
\geq 0. \quad (35)
$$

In this case, player $n$ may incur negative utility when it has type $t_n$ with $\theta(t_n) < 0$, although the loss is guaranteed to be bounded by $|\theta(t_n)|$.

### A.2.2 ON DEPENDENT TYPES

The condition (13) may be satisfied even when types are dependent, and the optimality of our analytic solutions in Lemma 3 is guaranteed as long as (13) is satisfied. When types are dependent, however, there are cases where feasible solutions exist even when (13) is violated (Proposition 1).

In the proof of Proposition 1, we construct such a case with an extreme example of completely dependent types. However, (13) is often satisfied even in such extreme cases of completely dependent types. For example, as long as

$$
x_1 \leq x_2 \leq \frac{2-p}{1-p} x_1, \quad (36)
$$

condition (13) is satisfied in the example in the proof of Proposition 1, since then $(x_1, x_2)$ satisfies (63), which corresponds to (13) in this example.

### A.3 DETAILS OF SECTION 6

### A.3.1 UPPER BOUND

For effective use of sample, we consider Algorithm 2, which wraps an arbitrary $(\varepsilon, \delta)$-PAC BAI. Specifically, Algorithm 2 not only returns the arm $\hat{I}$ that is identified as best by the $(\varepsilon, \delta)$-PAC BAI but also returns the number $M_{\hat{I}}$ of i.i.d. samples that the $(\varepsilon, \delta)$-PAC BAI has taken from $\hat{I}$ as well as the corresponding sample average $\mu_{\hat{I}}$. However, if no such information is available, it is perfectly fine, and Algorithm 2 simply returns $M_{\hat{I}} = 0$ and $\mu_{\hat{I}} = 0$ in such a case.

Notice that the naive approach of sampling each arm $\Theta((1/\varepsilon^2) \log(1/\delta))$ times, which also trivially falls within the upper bound in Theorem 1, would only guarantee that the best mean is estimated

with the error bound of $\varepsilon$ with probability at least $(1 - \delta)^K$. Conversely, it would require sampling each arm $\Omega((1/\varepsilon^2) \log(K/\delta))$ times to obtain the same error bound with probability $1 - \delta$.

### A.3.2 Lower bound

The lower bound established in Lemma 4 reduces BME to the $\varepsilon$-Biased Coin Problem defined as follows:

**Definition 3** ($\varepsilon$-Biased Coin Problem). *For $0 < \varepsilon < 1$, consider a Bernoulli random variable $B$ whose mean $\alpha$ is known to be either $\alpha^+ := (1 + \varepsilon)/2$ or $\alpha^- := (1 - \varepsilon)/2$. The $\varepsilon$-Biased Coin Problem asks to correctly identify whether $\alpha = \alpha^+$ or $\alpha = \alpha^-$.*

A lower bound on the sample complexity for solving the $\varepsilon$-Biased Coin Problem is known as the following lemma, for which we provide a proof in Appendix B for completeness:

**Lemma 7** (Chernoff (1972); Even-Dar et al. (2002)). *For $0 < \delta < 1/2$, any algorithm that solves the $\varepsilon$-Biased Coin Problem correctly with probability at least $1 - \delta$ has expected sample complexity at least $\Omega((1/\varepsilon^2) \log(1/\delta))$.*

In Lemma 5, we have derived the lower bound for BME using the technique used for a lower bound for BAI in Even-Dar et al. (2002); however, as we have discussed at the end of Section 6, while our lower bound for BME is tight, the lower bound for BAI in Even-Dar et al. (2002) is not. The difference in the derived lower bound stems from the following behavior of BME and BAI when all of the arms have mean reward of $\alpha^-$ and hence are indistinguishable. The algorithm constructed in Even-Dar et al. (2002) determines that $B$ has mean $\alpha^+$ when either arm $i^+$ or arm $i^-$ is identified as the best arm. When the arms are indistinguishable, a PAC BAI would correctly identify each of the $K$ arms, including $i^+$ or $i^-$, as the best arm uniformly at random, which induces an error with probability $1/K$. On the other hand, the mean reward estimated by a PAC BME would be approximately correct with high probability, even when the arms are indistinguishable.

### A.4 Details of Section 7

In this section, we conduct several numerical experiments to validate the effectiveness of the proposed approach and to understand its limitations. Specifically, we design our experiments to investigate the following three questions:

1. BME studied in Section 6 has asymptotically optimal sample complexity, but how well can we estimate the best mean when there are only a moderate number of arms?

2. How much can BME reduce the number of times $w^\star(t)$ given by (7) is evaluated?

3. How well $\theta$-IR and $\rho$-SBB are satisfied when (20) is estimated by BME rather than calculated exactly?

### A.4.1 Best Mean Estimation with moderate number of arms

For BAI, existing algorithms that have asymptotically optimal sample complexity often perform poorly with moderate number of arms (Hassidim et al., 2020). As a result, Approximate Best Arm (Hassidim et al., 2020), which has optimal sample complexity, runs Naive Elimination, which has asymptotically suboptimal sample complexity, when the number of arms is below $10^5$ or after sufficient number of suboptimal arms is eliminated. The BME algorithm studied in Section 6 has asymptotically optimal sample complexity but relies on BAI, and thus its sample complexity for moderate number of arms is not well characterized by Theorem 1. Similar to BAI, BME relies on an algorithm that performs well when there are only moderate number of arms.

In this section, we compare the performance of Successive Elimination algorithms for BAI and BME. Successive Elimination has suboptimal sample complexity, similar to Naive Elimination, but often outperforms Naive Elimination for moderate number of arms (Even-Dar et al., 2006). Specifically, we compare the performance of $(\varepsilon, \delta)$-PAC Successive Elimination for BME (SE-BME; Algorithm 4) against the corresponding $(\varepsilon, \delta)$-PAC Successive Elimination for BAI (SE-BAI; Algorithm 5). For completeness, in Appendix B, we provide standard proofs on the correctness of these algorithms, as stated in Proposition 5 and Proposition 6:

**Algorithm 4** Successive Elimination for Best Mean Estimation (($\varepsilon, \delta$)-PAC SE-BME)

**Require:** $\varepsilon, \delta$
1: Let $\mathcal{R} \leftarrow \{1, \ldots, K\}$ be the set of remaining arms
2: Let $t \leftarrow 0; \alpha \leftarrow 1$
3: **while** $\alpha > \varepsilon$ **do**
4:     Pull each arm $k \in \mathcal{R}$ once and update the sample average $\hat{\mu}_k$
5:     $t \leftarrow t + 1$
6:     $\alpha \leftarrow \sqrt{\frac{1}{2t} \log\left(\frac{\pi^2 K t^2}{3\delta}\right)}$
7:     **for all** $k \in \mathcal{R}$ **do**
8:         Remove $k$ from $\mathcal{R}$ if $\max_{\ell \in \mathcal{R}} \hat{\mu}_\ell - \hat{\mu}_k \geq 2\alpha$
9:     **end for**
10: **end while**
11: **return** $\max_{k \in \mathcal{R}} \hat{\mu}_k$

---

**Algorithm 5** Successive Elimination for Best Arm Identification (($\varepsilon, \delta$)-PAC SE-BAI)

**Require:** $\varepsilon, \delta$
1: Let $\mathcal{R} \leftarrow \{1, \ldots, K\}$ be the set of remaining arms
2: Let $t \leftarrow 0; \alpha \leftarrow 1$
3: **while** $|\mathcal{R}| > 1$ and $\alpha > \frac{\varepsilon}{2}$ **do**
4:     Pull each arm $k \in \mathcal{R}$ once and update the sample average $\hat{\mu}_k$
5:     $t \leftarrow t + 1$
6:     $\alpha \leftarrow \sqrt{\frac{1}{2t} \log\left(\frac{\pi^2 K t^2}{6\delta}\right)}$
7:     **for all** $k \in \mathcal{R}$ **do**
8:         Remove $k$ from $\mathcal{R}$ if $\max_{\ell \in \mathcal{R}} \hat{\mu}_\ell - \hat{\mu}_k \geq 2\alpha$
9:     **end for**
10: **end while**
11: **return** $\underset{k \in \mathcal{R}}{\operatorname{argmax}} \hat{\mu}_k$

---

**Proposition 5.** *Algorithm 4 is an $(\varepsilon, \delta)$-PAC BME.*

**Proposition 6.** *Algorithm 5 is an $(\varepsilon, \delta)$-PAC BAI.*

There are three differences between SE-BME and SE-BAI. First, SE-BME exits the while-loop when $\alpha < \varepsilon$ (while SE-BAI needs to wait until $\alpha < \varepsilon/2$), since it can return an overestimated mean of a suboptimal arm as long as the estimated value is within $\varepsilon$ from the best mean. Second, SE-BAI exits the while-loop when only one arm remains, since it does not need to precisely estimate the mean of the identified arm. Finally, SE-BAI uses smaller $\alpha_t$, since it only need to consider one-sided estimation error (underestimation for best arms and overestimation for other arms).

Figure 3 shows the total sample size required by SE-BME and by SE-BAI for varying values of $\varepsilon$ and $\delta$, and for varying number $K$ of arms. Here, the arms have Bernoulli rewards, and their means are selected in a way they are equally separated (i.e., $\mu_k = (k - 0.5)/K$ for $k = 1, \ldots, K$). For each datapoint, the experiments are repeated 10 times. The standard deviation of the total sample size is plotted but too small to be visible in the figure.

Overall, it can be observed that SE-BME generally requires smaller sample size than SE-BAI, except when there are only a few arms (and the means have large gaps in the setting under consideration). The efficiency of SE-BAI for a small number of arms makes intuitive sense, because the best arm can be identified without estimating the means with high accuracy.

A.4.2    EFFECTIVENESS OF BEST MEAN ESTIMATION IN MECHANISM DESIGN

In this section, we quantitatively validate the effectiveness of BME in reducing the number of times $w^\star(t)$ is computed when we evaluate (20). We use $(\varepsilon, \delta)$-PAC SE-BME (Algorithm 4) as BME.

To this end, we consider the following mechanism design, motivated by the double-sided auctions for electricity (Zou, 2009; Hobbs et al., 2000), where we have $N = |\mathcal{N}|$ players, each player $n \in \mathcal{N}$

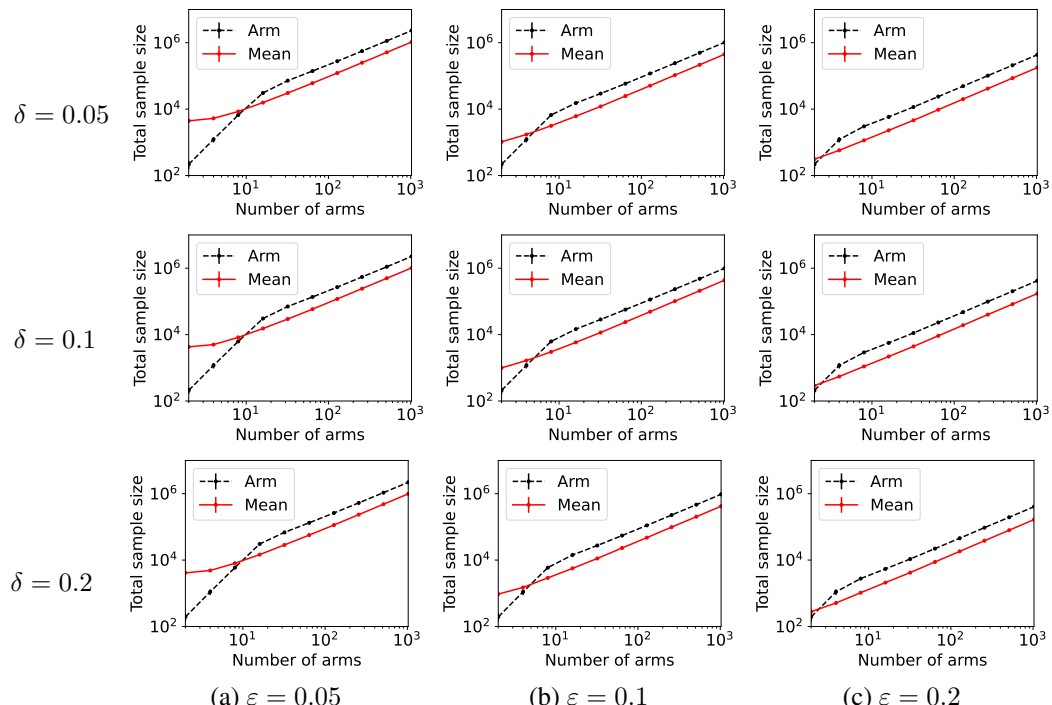

(a) $\varepsilon = 0.05$      (b) $\varepsilon = 0.1$      (c) $\varepsilon = 0.2$

Figure 3: The total sample size required by $(\varepsilon, \delta)$-PAC BME (Mean; Algorithm 4) and by $(\varepsilon, \delta)$-PAC BAI (Arm; Algorithm 5) when arms have Bernoulli rewards with equally separated means for varying values of $\varepsilon$ and $\delta$.

has $K$ possible types (i.e., $|\mathcal{T}_n| = K$), with varying values of $N$ and $K$. The $K$ possible types of each player are selected uniformly at random from integers, $[-K, K]$, without replacement. We then assume, as the common prior $\mathbb{P}$, that the type of each player is distributed uniformly among the $K$ possible types and independent of the types of other players. Each player is a buyer or a seller of a single item, depending on its type. When a player $n$ has a positive type $t_n$, the player is a buyer who wants to buy a unit, whose valuation to the player is $t_n$ (i.e., $v(d; t_n) = t_n$ if player $n$ buys a unit of the item with social decision $d$). When the player has a negative type $t_n$, the player is a seller who wants to sell a unit, which incurs cost $|t_n|$ (i.e., $v(d; t_n) = t_n$ if player $n$ sells a unit of the item with social decision $d$). The player does not participate in the market, when its type is zero. For a given profile of types $t$, a social decision is given by a bipartite matching between buyers and sellers. In this setting, we can immediately obtain the efficient social decision (3) by greedily matching buyers of high valuations to sellers of low costs as long as the value of the buyers are higher than the costs of the sellers.

Figure 4 shows representative sample paths of the estimated values of $\mathbb{E}[w^\star(t) \mid t_n]$ for each $t_n \in \mathcal{T}_n$ when $(0.25, 0.1)$-PAC SE-BME[1] is used to evaluate $\min_{t_n \in \mathcal{T}_n} \mathbb{E}[w^\star(t) \mid t_n]$ (i.e., (20) with $\theta(t_n) = 0$). Each panel in Figure 4 shows $|\mathcal{T}_n|$ curves, where the red curve corresponds to the one with minimum $\mathbb{E}[w^\star(t) \mid t_n]$.

Observe that the types $t_n$ (arms) that have close to the minimum mean, $\min_{t_n \in \mathcal{T}_n} \mathbb{E}[w^\star(t) \mid t_n]$, survive until SE-BME terminates, and their means are evaluated with sufficient accuracy. On the other hand, the types that have large means are eliminated after a relatively small number of samples without being estimated precisely, which contributes to reducing the number of evaluating the value of the efficient social decision $w^\star(t)$.

In Figure 5, we study how much SE-BME can reduce the number of evaluations of $w^\star(t)$ needed to estimate $\min_{t_n \in \mathcal{T}_n} \mathbb{E}[w^\star(t) \mid t]$ for all $n \in \mathcal{N}$ with the accuracy that is shown in Figure 4 (i.e., using the same values of $\varepsilon = 0.25$ and $\delta = 0.1$). Recall that $K = |\mathcal{T}_n|, \forall n \in \mathcal{N}$ in the setting under consideration. Hence, the exact computation of $\min_{t_n \in \mathcal{T}_n} \mathbb{E}[w^\star(t) \mid t_n]$ for a single $n \in \mathcal{N}$

---

[1]Minimization in Algorithm 4 is translated into maximization.

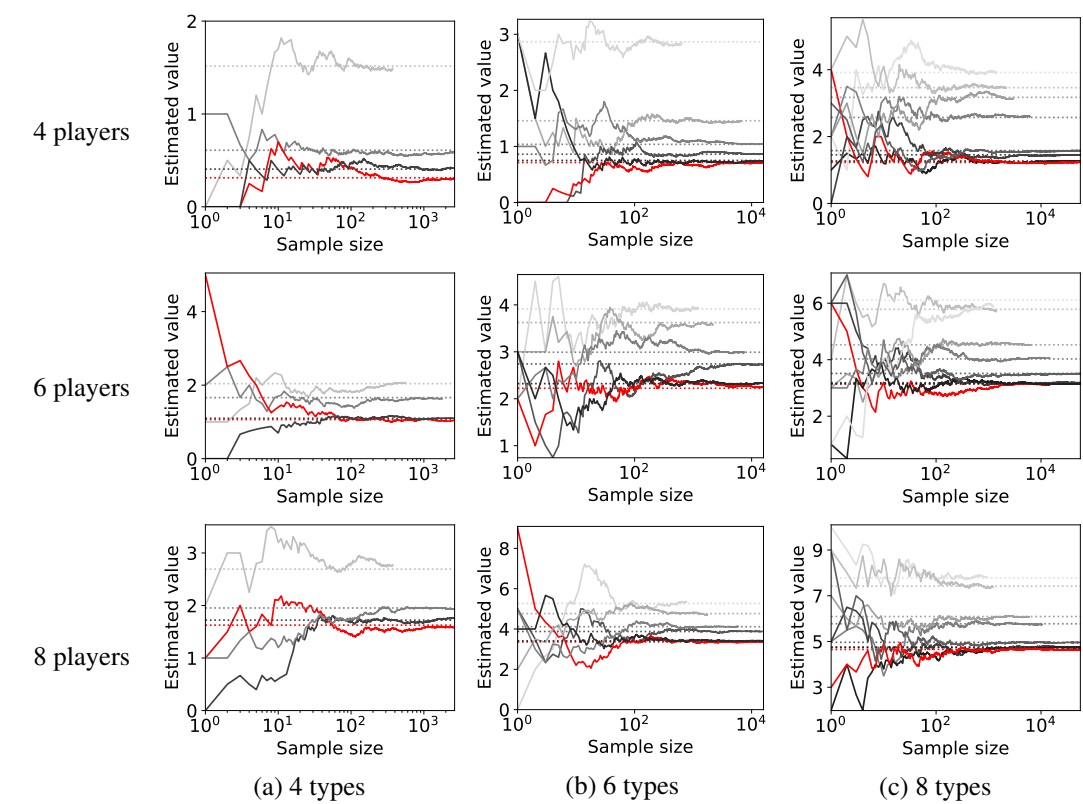

Figure 4: Representative sample paths that show the estimated values of $\mathbb{E}[w^\star(t) \mid t_n]$ for $t_n \in \mathcal{T}_n$ against sample size used by $(0.25, 0.1)$-PAC SE-BME.

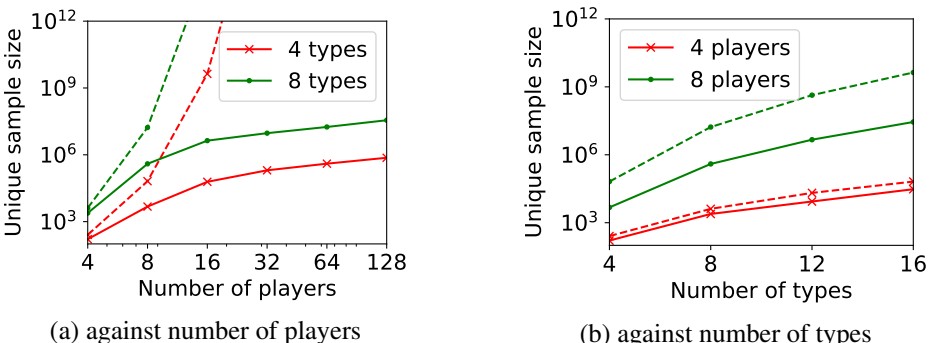

Figure 5: The unique sample size (the number of unique $t$ which $w^\star(t)$ is evaluated with) required by the exact computation of $\min_{t_n \in \mathcal{T}_n} \mathbb{E}[w^\star(t) \mid t_n], \forall n \in \mathcal{N}$ (dashed curves) and by $(0.25, 0.1)$-PAC SE BME (solid curves).

would require evaluating $w^\star(t)$ for $K^N$ different values of $t$, but $K^N$ evaluations of $w^\star(t)$ are also sufficient to exactly compute $\min_{t_n \in \mathcal{T}_n} \mathbb{E}[w^\star(t) \mid t_n]$ for all $n \in \mathcal{N}$, because we can cache the value of $w^\star(t)$ and reuse it when it is needed. Since the computational complexity associated with evaluating $w^\star(t)$ with (7) is the bottleneck, the unique sample size is what we should be interested in. Similar to exact computation, SE-BME also benefits from caching and reusing the values of $w^\star(t)$. Figure 5 compares the unique sample size required by the exact computation and $(0.25, 0.1)$-PAC SE-BME.

Figure 5(a) implies that SE-BME (shown with solid curves) evaluates the total value of the efficient social decision, $w^\star(t)$, by orders of magnitude smaller number of times than what is required by

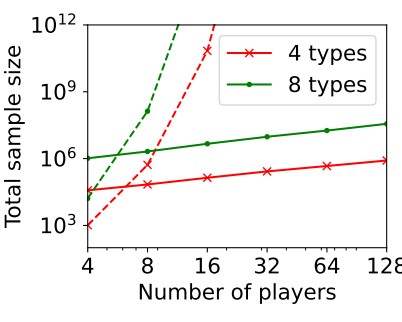

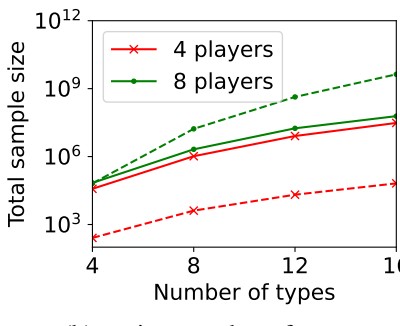

(a) against number of players

(b) against number of types

Figure 6: The total sample size (the number of $t$ which $w^\star(t)$ is evaluated with) required by the exact computation of $\min_{t_n \in \mathcal{T}_n} \mathbb{E}[w^\star(t) \mid t_n], \forall n \in \mathcal{N}$ (dashed curves) and by $(0.25, 0.1)$-PAC SE-BME (solid curves).

exact computation (shown with dashed curves). While the number of evaluations of $w^\star(t)$ grows exponentially with the number of players (specifically, $K^N$) when exact computation is used, it grows only polynomially (in fact, slightly slower than linearly) when SE-BME is used. This relative insensitivity of the sample complexity of SE-BME to the number of players makes intuitive sense, because the number of players only affects the distribution of the reward and keeps the number of arms unchanged.

Figure 5(b) shows the number of evaluations of $w^\star(t)$ against the number of types $K = |\mathcal{T}_n|$ for any $n \in \mathcal{N}$. The advantage of SE-BME over exact computation is relatively minor when we increase the number of types instead of the number of players, since increasing the number of types directly increases the number of arms. In all cases, however, we can observe that SE-BME can significantly reduce the unique sample size.

Figure 6 shows the total sample size, rather than the unique sample size, required by exact computation (dashed curves) and SE-BME (solid curves). The total sample size with exact computation is $N K^N$. While $w^\star(t)$ is evaluated $N$ times for each $t$ with exact computation, SE-BME may waste evaluating the same $w^\star(t)$ more often particularly when there are only a small number of players. This reduces benefits of SE-BME for total sample size, as compared to the unique sample size.

### A.4.3 INDIVIDUAL RATIONALITY AND BUDGET BALANCE WITH BEST MEAN ESTIMATION

We next address the question of how well $\theta$-IR and $\rho$-SBB are guaranteed when we estimate $\min_{t_n \in \mathcal{T}_n} \{\mathbb{E}[w^\star(t) \mid t_n] - \theta(t_n)\}$ with BME rather than computing it exactly. We continue to use the setting of mechanism design introduced in Section A.4.2. Recall that $\theta$-IR is guaranteed when $\eta_n$ is given by (57) and (18), and $\rho$-SBB is guaranteed when $\eta_n$ is given by (57) and (58). However, these are guaranteed only when the expected values, $\mathbb{E}[w^\star(t)]$ and $\mathbb{E}[w^\star(t) \mid t_n]$, are exactly computed. In this section, we quantitatively evaluate how well $\theta$-IR and $\rho$-SBB are satisfied when those expected values are estimated from samples. Throughout this section, we study the case with $\rho = 0$ and $\theta \equiv 0$ (i.e., $\theta(t_n) = 0, \forall t_n \in \mathcal{T}_n, \forall n \in \mathcal{N}$).

In Figure 7, we first evaluate the best mean, $\min_{t_n \in \mathcal{T}_n} \mathbb{E}[w^\star(t) \mid t]$, either with exact computation or with BME, then compute $\eta_n$ with (57) and (18) for Columns (a)-(b) and with (57) and (58) for Columns (c)-(d), and finally evaluate the expected utility of each player (the left-hand side of (11)) for Columns (a) and (c) and the expected revenue of the mediator (the left-hand side of (12)) for Columns (b) and (d) by setting $h_n(t_{-n}) = \eta_n, \forall t_{-n} \in \mathcal{T}_{-n}, \forall n \in \mathcal{N}$. Even if the best mean is estimated with BME, we evaluate the expectations on the left-hand side of (11) and (12) with exact computation, because these are the expected utility and the expected revenue that the players and the mediator will experience. Here, we repeat the experiment with 10 different random seeds, so that there are 10 data-points in Columns (b) and (d), and each of Columns (a) and (c) has $10 \times 8 \times 8 = 640$ data-points, where each data-point corresponds to a player of a particular type with a particular random seed.

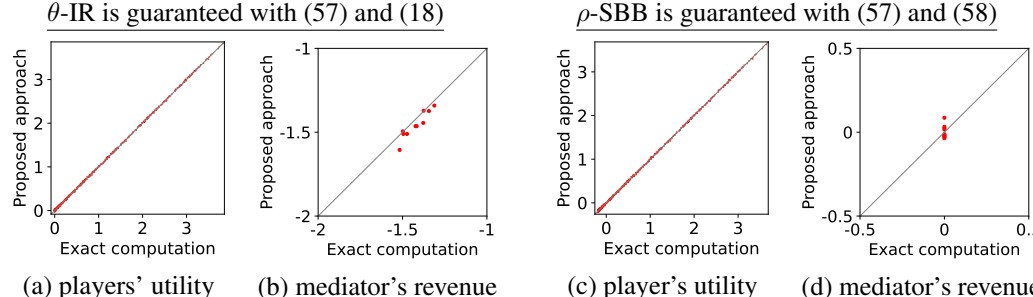

| (a) players' utility | (b) mediator's revenue | (c) player's utility | (d) mediator's revenue |

Figure 7: The red dots show the expected utility of the players in Columns (a) and (c) and the expected revenue of the mediator in Columns (b) and (d), where analytical solutions are evaluated exactly (horizontal axes) or estimated with $(0.25, 0.1)$-PAC SE-BME (vertical axes) for environments with $|\mathcal{N}| = 8$ players, each having $|\mathcal{T}_n| = 8$ possible types. The analytical solution guarantees $\theta$-IR with $\theta \equiv 0$ in Columns (a) and (b) and $\rho$-SBB with $\rho = 0$ in Columns (c) and (d). Results are plotted for 10 random seeds. Diagonal lines are also plotted to help understand where the horizontal and vertical axes are equal.

Overall, Columns (a) and (c) of Figure 7 show that the expected utility experienced by the players is relatively insensitive to whether the best mean is evaluated with exact computation (horizontal axes) or with BME (vertical axes). Taking a closer look, we can observe that, in this particular setting, 0-IR is violated for some of the players in Column (c) even if the best mean is computed exactly, while it is guaranteed for any player of any type in Column (a) if the best mean is computed exactly.

On the other hand, as is shown in Columns (b) and (d), the mediator experiences non-negligible difference in its expected revenue depending on whether the best mean is evaluated with exact computation or with BME. It is to be expected that the mediator experiences relatively larger variance in its expected revenue, because (12) involves the summation $\sum_{n \in \mathcal{N}} \eta_n$, while (11) only involves $\eta_n$ for a single $n \in \mathcal{N}$. In particular, it is evident in Column (d) that 0-WBB (let alone 0-SBB) is violated when the best mean is estimated with BME (vertical axes), while it is always guaranteed with exact computation (horizontal axes). In Column (b), 0-WBB is violated regardless of whether the best mean is computed exactly or with BME, since satisfying 0-WBB and 0-IR for all players (together with DE and DSIC) is impossible in this particular setting.

A simple remedy to this violation of $\rho$-WBB is to replace the $\rho$ with a $\rho' > \rho$ when we compute the $\eta_n$ from the best means, $\min_{t_n \in \mathcal{T}_n} \mathbb{E}[w^\star(t) \mid t_n]$ for $n \in \mathcal{N}$, estimated with BME. By considering the $(\varepsilon, \delta)$-PAC guarantee for the error in the estimation, we can guarantee that $\rho$-WBB is satisfied with high probability by setting an appropriate value of $\rho'$. Analogously, we may replace the $\theta(t_n)$ with a $\theta'(t_n) > \theta(t_n)$ to provide a guaranteed that $\theta$-IR is satisfied with high probability when the best mean is estimated with BME. See Theorem 2.

As an example, we set $\rho' = 0.1$ in Figure 8. The consequence of replacing $\rho = 0$ with $\rho' = 0.1$ is as expected. In Columns (b) and (d), the expected revenue of the mediator when the best mean is estimated with BME (Proposed approach) is shifted to the above by $\rho' - \rho = 0.1$. Although it may be unclear from Columns (a) and (c), the corresponding expected utility of each player is shifted to the left by $(\rho' - \rho)/|\mathcal{N}| = 0.0125$. In practice, we may choose $\rho'$ and $\theta'$ by taking into account these shifts as well as the condition on the feasibility of LP (Corollary 1).

In Figure 9, we show the Root Mean Squared Error (RMSE) in the expected utility of each player (a) and the expected revenue of the mediator (b) that are estimated with BME for the case with $|\mathcal{N}| = 8$ players, each with $|\mathcal{T}_n| = 4$ types. Here, we fix $\delta = 0.1$ and vary $\varepsilon$ from 1.0 to 0.15 in the BME. For each pair of $(\delta, \varepsilon)$, the experiments are repeated 10 times with different random seeds. The total sample size increases as the value of $\varepsilon$ decreases. Hence, the purple dots correspond to $\varepsilon = 1.0$, and the red dots are $\varepsilon = 0.15$. Overall, we can observe that RMSE can be reduced by using small $\varepsilon$ at the expense of increased sample size, that relatively large values such as $\varepsilon = 0.5$ gives reasonably small RMSE, and that larger values of $\varepsilon$ have diminishing effects on RMSE.

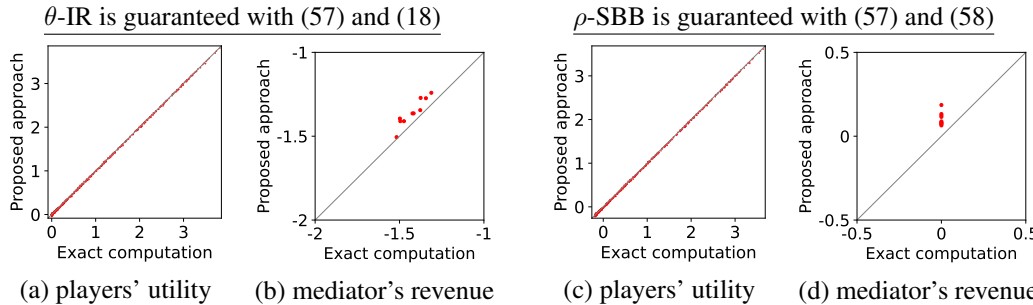

|  $\theta$-IR is guaranteed with (57) and (18) |  | $\rho$-SBB is guaranteed with (57) and (58) |  |
|---|---|---|---|
| (a) players' utility | (b) mediator's revenue | (c) players' utility | (d) mediator's revenue |

Figure 8: The red dots show the expected utility of the players in Columns (a) and (c) and the expected revenue of the mediator in Columns (b) and (d), where analytical solutions are evaluated exactly (horizontal axes) or estimated with $(0.25, 0.1)$-PAC SE-BME (vertical axes) for environments with $|\mathcal{N}| = 8$ players, each having $|\mathcal{T}_n| = 8$ possible types. The analytical solution guarantees $\theta$-IR with $\theta \equiv 0$ in Columns (a) and (b) and $\rho'$-SBB with $\rho' = 0.1$ in Columns (c) and (d). Results are plotted for 10 random seeds. Diagonal lines are also plotted to help understand where the horizontal and vertical axes are equal.

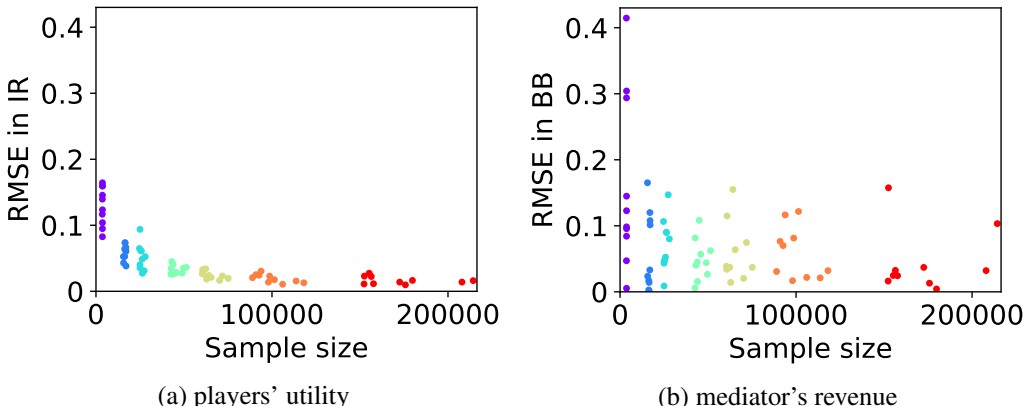

| (a) players' utility | (b) mediator's revenue |
|---|---|

Figure 9: Root mean squared error in (a) the expected utility of each player and (b) the expected revenue of the mediator against the total sample size, when there are $|\mathcal{N}| = 8$ players, each with $|\mathcal{T}_n| = 4$ possible types. Here, we set $\delta = 0.1$ and vary $\varepsilon$ from 1.0 (purple), 0.5, 0.4, 0.3, 0.25, 0.2, to 0.15 (red).

### A.4.4 COMPUTATIONAL REQUIREMENTS

We have run all of the experiments on a single core with at most 66 GB memory without GPUs in a cloud environment. The associated source code is submitted as a supplementary material and will be open-sourced upon acceptance. Table 1 summarizes the CPU time and maximum memory require to generate each figure. For example, CPU time for Figure 3(a) is the time to generate three panels in Column (a) of Figure 3. Note that the CPU time and maximum memory reported in Table 1 are not optimized and include time and memory for storing intermediate results and other processing for debugging purposes; these should be understood as the computational requirements to execute the source code as is.

Table 1: CPU time and maximum memory required to generate figures

| Figure | CPU Time (seconds) | Max Memory (GB) |
| --- | --- | --- |
| Figure 3(a) | 413.1 | $< 1$ |
| Figure 3(b) | 81.9 | $< 1$ |
| Figure 3(c) | 40.4 | $< 1$ |
| Figure 4(a)[†] | 0.7 | $< 1$ |
| Figure 4(b)[†] | 1.1 | $< 1$ |
| Figure 4(c)[†] | 80.1 | 1.9 |
| Figure 5(a) and Figure 6(a) | 7,883.0 | 65.5 |
| Figure 5(b) and Figure 6(b) | 2,935.2 | 17.0 |
| Figure 7(a)-(b) and Figure 8(a)-(b)[‡] | 17,531.6 | 1.6 |
| Figure 7(c)-(d) and Figure 8(c)-(d)[‡] | 17,416.3 | 1.6 |
| Figure 9 | 527.9 | $< 1$ |

[†] Figure 4 shows the results with one random seed, but here the CPU Time reports the average over 10 seeds, and Max Memory reports the maximum over 10 seeds.

[‡] Figure 8 could have been obtained by simply reusing and shifting Figure 7, but here the CPU time reports the time to generate the two figures without reuse.

## A.5 DETAILS OF SECTION 8

### A.5.1 LIMITATIONS

While the proposed approach makes major advancement in the field, it certainly has limitations. Here, we discuss four major limitations of this work as well as interesting directions of research motivated by those limitations.

First, when types are not independent between players, the sufficient condition in Lemma 1 may not be necessary (Proposition 1). This means that the LP may be feasible even when the condition in the lemma is violated, and our results do not provide optimal solutions for those cases. Further research is needed to understand exactly when the sufficient condition is also necessary. It is also important to develop efficient methods for solving the LP when the types are dependent.

Second, our mechanisms guarantee strong budget balance (SBB) and individual rationality (IR) in expectation with respect to the distribution of the players' types, but this does not guarantee that those properties are satisfied *ex post* (for any realization of the types). Although the satisfaction in expectation is often sufficient for risk-neutral decision makers, it is important to let the mediator and the participants aware that they may experience negative utilities even if their expected utilities are nonnegative. It would also be an interesting direction of research to extend the proposed approach towards achieving these properties *ex post*.

Finally, our experiments have considered environments with up to 128 players, each with at most 16 types. Although these are substantially larger than the environment studied in Osogami et al. (2023), they certainly do not cover the scale needed for all applications. While (10-100 times) larger environments could be handled with improved implementation and greater computational resources, essentially new ideas would be needed for substantially (over $10^3$ times) larger environments or continuous type space. It would be an interesting direction of research to identify and exploit structures of particular environments for designing scalable approaches of mechanism design for those environments.

### A.5.2 SOCIETAL IMPACTS

We expect that the proposed approach has several positive impacts on trading networks in particular and the society in general. In particular, the proposed approach enables mechanisms that can maximize the efficiency of a trading network and minimize the fees that the participants need to pay to the mediator. Also, the DSIC guaranteed by the proposed approach would make it more difficult for malicious participants to manipulate the outcome of a trading network.

On the other hand, the proposed approach might have negative impacts depending on where and how it is applied. For example, although the proposed approach guarantees individual rationality, some of the participants might get less benefits from the mechanism designed with our approach than other participants. This can happen, because maximizing the social welfare does not mean that all the participants are treated fairly. Before applying the mechanisms designed with the proposed approach, it is thus recommended to assess whether such fairness needs to be considered and to take any actions that mitigate the bias if needed.

## B  PROOFS

### B.1  PROOFS ON THE LEMMAS AND COROLLARIES IN SECTION 5

*Proof of Lemma 1.* We will show that

$$h_n(t_{-n}) = \eta_n := \min_{t_n \in \mathcal{T}_n} \{\mathbb{E}[w^\star(t) \mid t_n] - \theta(t_n)\} \qquad \forall t_{-n} \in \mathcal{T}_{-n} \tag{37}$$

is a feasible solution when (13) holds.

The $\theta$-IR (11) is satisfied with (37), because for any $n \in \mathcal{N}$ we have

$$\mathbb{E}[h_n(t_{-n}) \mid t_n] = \eta_n \tag{38}$$

$$= \min_{t_n' \in \mathcal{T}_n} \{\mathbb{E}[w^\star(t') \mid t_n'] - \theta(t_n')\} \tag{39}$$

$$\leq \mathbb{E}[w^\star(t) \mid t_n] - \theta(t_n). \tag{40}$$

The $\rho$-WBB (12) is satisfied with (37), because

$$\sum_{n \in \mathcal{N}} \mathbb{E}[h_n(t_{-n})] = \sum_{n \in \mathcal{N}} \eta_n \tag{41}$$

$$= \sum_{n \in \mathcal{N}} \min_{t_n \in \mathcal{T}_n} \{\mathbb{E}[w^\star(t) \mid t_n] - \theta(t_n)\} \tag{42}$$

$$\geq (N - 1)\mathbb{E}[w^\star(t)] + \rho, \tag{43}$$

where the inequality follows from (13).  □

*Proof of Lemma 2.* By independence, (11) is reduced to

$$\mathbb{E}[w^\star(t) \mid t_n] - \mathbb{E}[h_n(t_{-n})] \geq \theta(t_n) \qquad \forall t_n \in \mathcal{T}_n, \forall n \in \mathcal{N} \tag{44}$$

$$\iff \min_{t_n \in \mathcal{T}_n} \{\mathbb{E}[w^\star(t) \mid t_n] - \theta(t_n)\} \geq \mathbb{E}[h_n(t_{-n})] \qquad \forall n \in \mathcal{N}. \tag{45}$$

This together with (12) establishes the necessity of

$$(N - 1)\mathbb{E}[w^\star(t)] + \rho \leq \sum_{n \in \mathcal{N}} \mathbb{E}[h_n(t_{-n})] \tag{46}$$

$$\leq \sum_{n \in \mathcal{N}} \min_{t_n \in \mathcal{T}_n} \{\mathbb{E}[w^\star(t) \mid t_n] - \theta(t_n)\}. \tag{47}$$

□

*Proof of Lemma 3.* We first rewrite the LP (10)-(12) in the following equivalent form:

$$\min_h \quad \sum_{n \in \mathcal{N}} \sum_{t_n \in \mathcal{T}_n} \mathbb{P}[t_n]\mathbb{E}[h_n(t_{-n}) \mid t_n] \tag{48}$$

$$\text{s.t.} \quad \mathbb{E}[w^\star(t) \mid t_n] - \mathbb{E}[h_n(t_{-n}) \mid t_n] \geq \theta(t_n) \qquad \forall t_n \in \mathcal{T}_n, \forall n \in \mathcal{N} \tag{49}$$

$$\sum_{n \in \mathcal{N}} \sum_{t_n \in \mathcal{T}_n} \mathbb{P}[t_n]\mathbb{E}[h_n(t_{-n}) \mid t_n] - (N - 1)\mathbb{E}[w^\star(t)] \geq \rho. \tag{50}$$

Then it can be easily observed that the optimal objective value must be equal to $(N-1)\,\mathbb{E}[w^\star(t)]+\rho$ (i.e., when equality holds in (50)), since changing $h$ in a way it decreases the value of $\mathbb{E}[h_n(t_{-n}) \mid t_n]$ only makes (49) more satisfiable[2].

Hence, to prove that any $h \in \mathcal{H}$ is an optimal solution, it suffices to show that (50) is satisfied with equality and (49) is satisfied with any $h \in \mathcal{H}$. When $h \in \mathcal{H}$, we have, for any $t_n$, that

$$\mathbb{E}[h_n(t_{-n}) \mid t_n] = \eta_n \tag{51}$$

$$\leq \min_{t'_n \in \mathcal{T}_n} \left\{ \mathbb{E}[w^\star(t) \mid t'_n] - \theta(t'_n) \right\} \tag{52}$$

$$\leq \mathbb{E}[w^\star(t) \mid t_n] - \theta(t_n), \tag{53}$$

where the first inequality follows from (16). We also have

$$\sum_{n \in \mathcal{N}} \sum_{t_n \in \mathcal{T}_n} \mathbb{P}[t_n]\,\mathbb{E}[h_n(t_{-n}) \mid t_n] = \sum_{n \in \mathcal{N}} \sum_{t_n \in \mathcal{T}_n} \mathbb{P}[t_n]\,\eta_n \tag{54}$$

$$= \sum_{n \in \mathcal{N}} \eta_n \tag{55}$$

$$= (N-1)\,\mathbb{E}[w^\star(t)] + \rho, \tag{56}$$

where the last equality follows from (15) and (17).

Finally, when (13) holds, $\mathcal{H}$ is nonempty, because the following $\eta_n$ satisfies the conditions (15)-(17):

$$\eta_n = \min_{t_n \in \mathcal{T}_n} \left\{ \mathbb{E}[w^\star(t) \mid t_n] - \theta(t_n) \right\} - \delta \tag{57}$$

where

$$\delta := \frac{1}{N}\left( \sum_{n \in \mathcal{N}} \min_{t_n \in \mathcal{T}_n} \left\{ \mathbb{E}[w^\star(t) \mid t_n] - \theta(t_n) \right\} - (N-1)\,\mathbb{E}[w^\star(t)] - \rho \right). \tag{58}$$

Notice that $\delta \geq 0$ follows from (13). $\qquad\square$

*Proof of Proposition 1.* We construct an example that satisfies (11)-(12) but violates (13). Let $\mathcal{N} = \{1,2\}$; $\mathcal{T}_n = \mathcal{T} := \{1,2\}, \forall n \in \mathcal{N}$; $\rho = 0$; $\theta(m) = 0, \forall m \in \mathcal{T}$. We assume that the types are completely dependent (namely, $t_1 = t_2$ surely) and let $p$ be the probability that $t_1 = t_2 = 1$ (hence, $t_1 = t_2 = 2$ with probability $1 - p$).

For this example, we rewrite (11)-(12) and (13) by using $x_m := w^\star((m,m))$ and $y_{nm} := h_n(m)$ for $m \in \mathcal{T}$ and $n \in \mathcal{N}$. Notice that, for any $m \in \mathcal{T}$ and $n \in \mathcal{N}$, we have

$$\mathbb{E}[w^\star(t) \mid t_n = m] = x_m \tag{59}$$

$$\mathbb{E}[h_n(t_{-n}) \mid t_n = m] = y_{nm}, \tag{60}$$

since types are completely dependent. Hence, (11) is reduced to

$$x_m - y_{nm} \geq 0 \qquad \forall m \in \mathcal{T}, \forall n \in \mathcal{N} \tag{61}$$

and (12) is reduced to

$$p\,(y_{11} + y_{21} - x_1) + (1-p)\,(y_{12} + y_{22} - x_2) \geq 0. \tag{62}$$

On the other hand, (13) is reduced to

$$2\,\min\{x_1, x_2\} \geq p\,x_1 + (1-p)\,x_2. \tag{63}$$

Consider the case where $x_m > 0, \forall m \in \mathcal{T}$. In this case, (61)-(62) suggest that (11)-(12) are satisfied as long as $y_{nm}$ satisfies

$$\frac{x_m}{2} \leq y_{nm} \leq x_m \qquad \forall m \in \mathcal{T}, \forall n \in \mathcal{N}, \tag{64}$$

whether (63) is satisfied or not. Indeed, (64) can be met even if (63) is violated, for example when $p = \frac{1}{2}, x_1 = 1, x_2 = 4, y_{nm} = \frac{2}{3}x_m, \forall m \in \mathcal{T}, \forall n \in \mathcal{N}$; this serves as an desired example, concluding the proof. $\qquad\square$

---

[2]This implies that $\rho$-SBB is satisfied whenever $\rho$-WBB is satisfied

*Proof of Corollary 2.* The sufficiency follows from Lemma 3. The necessity follows in exactly the same way as the proof of Lemma 2. □

*Proof of Corollary 3.* By (15), we have

$$\sum_{n \in \mathcal{N}} \eta_n - (N-1)\,\mathbb{E}[w^\star(t)] - \rho$$

$$= \sum_{n \in \mathcal{N}} \min_{t_n \in \mathcal{T}_n} \left\{ \mathbb{E}[w^\star(t) \mid t_n] - \theta(t_n) \right\} - \sum_{n \in \mathcal{N}} \delta_n - (N-1)\,\mathbb{E}[w^\star(t)] - \rho \tag{65}$$

$$= 0, \tag{66}$$

where the last equality follows from (17). Hence, (12) is satisfied with equality. □

*Proof of Corollary 4.* This corollary can be proved analogously to Corollary 3. □

*Proof of Corollary 5.* By (15), for any $t_n \in \mathcal{T}_n$ and $n \in \mathcal{N}$, we have

$$\mathbb{E}[w^\star(t) \mid t_n] - \eta_n - \theta(t_n)$$

$$= \mathbb{E}[w^\star(t) \mid t_n] - \theta(t_n) - \left( \min_{t'_n \in \mathcal{T}_n} \left\{ \mathbb{E}[w^\star(t) \mid t'_n] - \theta(t'_n) \right\} \right) + \delta_n, \tag{67}$$

which is nonnegative by (16), and hence (11) holds. □

### B.2 Proofs of the lemmas, theorem, and proposition in Section 6

*Proof of Lemma 4.* Since the sample complexity of PAC-BAI in Step 1 is $M$ and Step 3 pulls an arm at most $m^\star$ times, the sample complexity of Algorithm 3 is at most $M + m^\star$. Hence, it remains to prove (23).

Recall that $\hat{I}$ is a random variable representing the index of the best arm returned by Algorithm 2 (PAC-BAI). Then we have the following bound:

$$\Pr \left( |\hat{\mu}_{\hat{I}} - \mu_\star| > \frac{3}{2}\varepsilon \right)$$

$$= \Pr \left( \hat{\mu}_{\hat{I}} > \mu_\star + \frac{3}{2}\varepsilon \right) + \Pr \left( \hat{\mu}_{\hat{I}} < \mu_\star - \frac{3}{2}\varepsilon \right) \tag{68}$$

$$\leq \Pr \left( \hat{\mu}_{\hat{I}} > \mu_{\hat{I}} + \frac{3}{2}\varepsilon \right)$$

$$\quad + \Pr \left( \left\{ \hat{\mu}_{\hat{I}} < \mu_\star - \frac{3}{2}\varepsilon \right\} \cap \left\{ \mu_{\hat{I}} < \mu_\star - \varepsilon \right\} \right) + \Pr \left( \left\{ \hat{\mu}_{\hat{I}} < \mu_\star - \frac{3}{2}\varepsilon \right\} \cap \left\{ \mu_{\hat{I}} \geq \mu_\star - \varepsilon \right\} \right) \tag{69}$$

$$\leq \Pr \left( \hat{\mu}_{\hat{I}} > \mu_{\hat{I}} + \frac{3}{2}\varepsilon \right) + \Pr \left( \mu_{\hat{I}} < \mu_\star - \varepsilon \right) + \Pr \left( \left\{ \hat{\mu}_{\hat{I}} < \mu_\star - \frac{3}{2}\varepsilon \right\} \cap \left\{ \mu_{\hat{I}} \geq \mu_\star - \varepsilon \right\} \right) \tag{70}$$

$$\leq \Pr \left( \hat{\mu}_{\hat{I}} > \mu_{\hat{I}} + \frac{3}{2}\varepsilon \right) + \Pr \left( \mu_{\hat{I}} < \mu_\star - \varepsilon \right) + \Pr \left( \hat{\mu}_{\hat{I}} < \mu_{\hat{I}} - \frac{1}{2}\varepsilon \right) \tag{71}$$

$$\leq \Pr \left( \hat{\mu}_{\hat{I}} > \mu_{\hat{I}} + \frac{3}{2}\varepsilon \right) + \delta + \Pr \left( \hat{\mu}_{\hat{I}} < \mu_{\hat{I}} - \frac{1}{2}\varepsilon \right), \tag{72}$$

where the last inequality follows from PAC($\varepsilon, \delta$) of BAI.

Since $\hat{\mu}_{\hat{I}}$ is the average of $M' \coloneqq \max\{M_{\hat{I}}, m^\star\}$ samples from arm $\hat{I}$, where $M_{\hat{I}}$ is the number of times Algorithm 2 (PAC-BAI) has pulled arm $\hat{I}$, we have $M' \geq m^\star$. Hence, applying Hoeffding's

inequality to the last term of (72), we obtain

$$\Pr\left(\hat{\mu}_{\hat{I}} < \mu_{\hat{I}} - \frac{1}{2}\varepsilon\right) = \sum_{n \geq m^{\star}, k \in [1,K]} \Pr\left(\hat{\mu}_k < \mu_k - \frac{1}{2}\varepsilon \,\middle|\, M' = n, \hat{I} = k\right) \Pr(M' = n, \hat{I} = k) \tag{73}$$

$$\leq \sum_{n \geq m^{\star}, k \in [1,K]} \exp\left(-2\left(\frac{1}{2}\varepsilon\right)^2 n\right) \Pr(M' = n, \hat{I} = k) \tag{74}$$

$$\leq \exp\left(-2\left(\frac{1}{2}\varepsilon\right)^2 m^{\star}\right), \tag{75}$$

where the first inequality is obtained by applying Hoeffding's inequality to the sample mean $\hat{\mu}_k$ of $n$ independent random variables having support in $[0,1]$, and the second inequality follows from $n \geq m^{\star}$. We can also show the following inequality in an analogous manner:

$$\Pr\left(\hat{\mu}_{\hat{I}} > \mu_{\hat{I}} + \frac{3}{2}\varepsilon\right) \leq \exp\left(-2\left(\frac{3}{2}\varepsilon\right)^2 m^{\star}\right). \tag{76}$$

By applying (75)-(76) to (72), we finally establish the bound to be shown:

$$\Pr\left(|\hat{\mu}_{\hat{I}} - \mu_{\star}| > \frac{3}{2}\varepsilon\right) \leq \delta + \exp\left(-2\left(\frac{3}{2}\varepsilon\right)^2 m^{\star}\right) + \exp\left(-2\left(\frac{1}{2}\varepsilon\right)^2 m^{\star}\right) \tag{77}$$

$$\leq \delta + \exp\left(-2\left(\frac{3}{2}\varepsilon\right)^2 \frac{2}{\varepsilon^2}\log\frac{1.22}{\delta}\right) + \exp\left(-2\left(\frac{1}{2}\varepsilon\right)^2 \frac{2}{\varepsilon^2}\log\frac{1.22}{\delta}\right)$$

$$\text{by the definition of } m^{\star} \tag{78}$$

$$= \delta + \left(\frac{\delta}{1.22}\right)^9 + \frac{\delta}{1.22} \tag{79}$$

$$\leq \left(1 + \frac{1}{1.22^9} + \frac{1}{1.22}\right)\delta \tag{80}$$

$$\leq 2\,\delta. \tag{81}$$

$\square$

*Proof of Lemma 7.* Although the lemma is stated in Even-Dar et al. (2002) with reference to Chernoff (1972), this specific lemma is neither stated nor proved explicitly in Chernoff (1972). For completeness, here, we prove the lemma following the general methodology provided in Chernoff (1972). Specifically, we derive the expected sample size required by the sequential probability-ratio test (SPRT; Section 10 of Chernoff (1972)), whose optimality (Theorem 12.1 of Chernoff (1972)) will then establish the lemma.

Consider two hypotheses, $\theta_1$ and $\theta_2$, for the probability distribution $\mathbb{P}$ of a random variable $X$, which takes either the value of 1 or $-1$, where

$$\mathbb{P}(X = 1 \mid \theta_1) = \frac{1 + \varepsilon}{2} \tag{82}$$

$$\mathbb{P}(X = 1 \mid \theta_2) = \frac{1 - \varepsilon}{2} \tag{83}$$

Consider the SPRT procedure that takes i.i.d. samples, $X_1, X_2, \ldots, X_N$, from $\mathbb{P}$ until the stopping time $N$ when

$$\lambda_N := \prod_{n=1}^{N} \frac{\mathbb{P}(X_n \mid \theta_1)}{\mathbb{P}(X_n \mid \theta_2)} = \prod_{n=1}^{N} \left(\frac{\varepsilon + 1}{\varepsilon - 1}\right)^{X_n} \tag{84}$$

hits either $A \in \mathbb{R}$ or $1/A$. When $\lambda_N$ hits $A$, we identify $\theta_1$ as the correct hypothesis. When $\lambda_N$ hits $1/A$, we identify $\theta_2$ as the correct hypothesis.

Let

$$S_N := \log \lambda_N = \sum_{n=1}^{N} X_n \, \log \frac{1+\varepsilon}{1-\varepsilon}. \tag{85}$$

Since $N$ is a stopping time, by Wald's lemma, we have

$$\mathbb{E}[S_N \mid \theta_1] = \mathbb{E}[N \mid \theta_1] \, \mathbb{E}[X \mid \theta_1] \, \log \frac{1+\varepsilon}{1-\varepsilon} \tag{86}$$

$$= \mathbb{E}[N \mid \theta_1] \, \varepsilon \, \log \frac{1+\varepsilon}{1-\varepsilon} \tag{87}$$

$$\mathbb{E}[S_N \mid \theta_2] = -\mathbb{E}[N \mid \theta_2] \, \varepsilon \, \log \frac{1+\varepsilon}{1-\varepsilon}. \tag{88}$$

Let $\delta$ be the probability of making the error in identifying the correct hypothesis. Then we must have

$$\mathbb{E}[S_N \mid \theta_1] = (1-\delta) \log A + \delta \log(1/A) \tag{89}$$
$$\mathbb{E}[S_N \mid \theta_2] = \delta \log A + (1-\delta) \log(1/A). \tag{90}$$

By (86)-(90), we have

$$\mathbb{E}[N \mid \theta_1] = \mathbb{E}[N \mid \theta_2] = \frac{1-2\delta}{\varepsilon \log \frac{1+\varepsilon}{1-\varepsilon}} \, \log A. \tag{91}$$

Now, notice that $S_N$ hits $\log A$ when we have

$$|\{n : X_n = 1\}| - |\{n : X_n = -1\}| \geq \frac{\log A}{\log \frac{1+\varepsilon}{1-\varepsilon}} \tag{92}$$

for the first time and hits $-\log A$ when we have

$$|\{n : X_n = -1\}| - |\{n : X_n = 1\}| \geq \frac{\log A}{\log \frac{1+\varepsilon}{1-\varepsilon}} \tag{93}$$

for the first time. Hence, by the gambler's ruin probability, we have

$$\delta = \frac{1 - \left(\frac{1+\varepsilon}{1-\varepsilon}\right)^{\frac{\log A}{\log \frac{1+\varepsilon}{1-\varepsilon}}}}{1 - \left(\frac{1+\varepsilon}{1-\varepsilon}\right)^{2\frac{\log A}{\log \frac{1+\varepsilon}{1-\varepsilon}}}} = \frac{1}{1 + \left(\frac{1+\varepsilon}{1-\varepsilon}\right)^{\frac{\log A}{\log \frac{1+\varepsilon}{1-\varepsilon}}}}, \tag{94}$$

which implies

$$A = \frac{1-\delta}{\delta}. \tag{95}$$

Plugging the last expression into (91), we obtain

$$\mathbb{E}[N \mid \theta_1] = \mathbb{E}[N \mid \theta_2] = \frac{1-2\delta}{\varepsilon \log \frac{1+\varepsilon}{1-\varepsilon}} \, \log \frac{1-\delta}{\delta} = O\left(\frac{1}{\varepsilon^2} \log \frac{1}{\delta}\right). \tag{96}$$

$$\square$$

*Proof of Lemma 5.* We will construct an algorithm that correctly identifies the mean $\alpha$ of $B$ with probability at least $\delta$ using at most $M/K$ samples of $B$ in expectation given the access to an $(\varepsilon/2, \delta/2)$-PAC BME with sample complexity $M$. The algorithm first draws $i^+$ and $i^-$ independently from a uniformly distribution over $[1, K]$.

Consider two environments of $K$-armed bandit, $\mathcal{E}^+$ and $\mathcal{E}^-$, where every arm gives the reward according to the Bernoulli distribution with mean $\alpha^- = (1-\varepsilon)/2$ except that the reward of arm $i^+$ in $\mathcal{E}^+$ has the same distribution as $B$ and the reward of arm $i^-$ in $\mathcal{E}^-$ has the same distribution as $1 - B$. Note that the algorithm can simulate the reward with the known mean $\alpha^-$ using the

algorithm's internal random number generators. Only when the algorithm pulls arm $i^+$ of $\mathcal{E}^+$ or arm $i^-$ of $\mathcal{E}^-$, it uses the sample of $B$, which contributes to the sample complexity of the algorithm.

The algorithm then runs two copies of the $(\varepsilon/2, \delta/2)$-PAC BME with sample complexity $M$ in parallel: one referred to as BME$^+$ is run on $\mathcal{T}^+$, and the other referred to as BME$^-$ is run on $\mathcal{T}^-$. At each step, let $M^+$ be the number of samples BME$^+$ has taken from arm $i^+$ by that step and $M^-$ be the corresponding number BME$^-$ has taken from arm $i^-$. If $M^+ < M^-$, the algorithm lets BME$^+$ pull an arm; otherwise, the algorithm lets BME$^-$ pull an arm. Therefore, $|M^+ - M^-| \leq 1$ at any step.

This process is continued until one of BME$^+$ and BME$^-$ terminates and returns an estimate $\hat{\mu}$ of the best mean. If BME$^+$ terminates first, then the algorithm determines that $\alpha = \alpha^-$ if $\hat{\mu} < 1/2$ and that $\alpha = \alpha^+$ otherwise. If BME$^-$ terminates first, then the algorithm determines that $\alpha = \alpha^+$ if $\hat{\mu} < 1/2$ and that $\alpha = \alpha^-$ otherwise. Due to the $(\varepsilon/2, \delta/2)$-PAC property of BME$^+$ and BME$^-$, the algorithm correctly identifies the mean of $B$ with probability at least $1 - \delta$. Formally, if $\alpha = \alpha^+$, we have

$$\Pr\left(\hat{\mu} < \frac{1}{2}\right) = \Pr\left(\hat{\mu} < \frac{1}{2} \,\middle|\, \text{BME}^+ \text{ terminates first}\right) \Pr(\text{BME}^+ \text{ terminates first})$$

$$+ \Pr\left(\hat{\mu} < \frac{1}{2} \,\middle|\, \text{BME}^- \text{ terminates first}\right) \Pr(\text{BME}^- \text{ terminates first}) \tag{97}$$

$$\leq \frac{\delta}{2} + \frac{\delta}{2} \tag{98}$$

$$= \delta. \tag{99}$$

Analogously, $\Pr\left(\hat{\mu} < \frac{1}{2}\right) \leq \delta$ can be shown if $\alpha = \alpha^-$.

What remains to prove is the sample complexity of the algorithm. Recall that each of BME$^+$ and BME$^-$ pulls arms at most $M$ times before it terminates due to their sample complexity. Notice that the arms in $\mathcal{E}^-$ are indistinguishable when $\alpha = \alpha^-$, and the arms in $\mathcal{E}^+$ are indistinguishable when $\alpha = \alpha^+$. Therefore, at least one of BME$^+$ and BME$^-$ is run on the environment where the arms are indistinguishable. Since $i^-$ and $i^-$ are sampled uniformly at random from $[1, K]$, BME (either BME$^+$ or BME$^-$) would take at most $M/n$ samples from $B$ in expectation if the arms are indistinguishable. Since $|M^+ - M^-| \leq 1$, we establish that the sample complexity of the algorithm is $O(M/n)$ in expectation. $\qquad\square$

*Proof of Lemma 6.* Note that the PAC estimators can give independent estimates, $\tilde{\kappa}_n(\theta)$ for $n \in \mathcal{N}$ and $\tilde{\lambda}$, such that

$$\Pr\left(|\tilde{\kappa}_n(\theta) - \kappa_n(\theta)| \leq \varepsilon'\right) \geq 1 - \delta' \qquad \forall n \in \mathcal{N} \tag{100}$$

$$\Pr\left(|\tilde{\lambda}(\theta) - \lambda(\theta)| \leq \varepsilon''\right) \geq 1 - \delta', \tag{101}$$

which imply

$$\Pr\left(|\tilde{\lambda}(\theta) - \lambda(\theta)| \leq \varepsilon'', |\tilde{\kappa}_n(\theta) - \kappa_n(\theta)| \leq \varepsilon', \forall n \in \mathcal{N}\right) \geq (1 - \delta')^{N+1}. \tag{102}$$

Hence, with probability at least $(1 - \delta')^{N+1}$, the expected utility of player $n$ given its type $t_n$ (recall (8)) is

$$\mathbb{E}[w^\star(t) \mid t_n] - (\tilde{\kappa}_n(\theta) - \tilde{d}_n) \geq \mathbb{E}[w^\star(t) \mid t_n] - \kappa_n(\theta) - \varepsilon' + \varepsilon''' \tag{103}$$

$$\geq \theta(t_n) - (\varepsilon' - \varepsilon'''), \tag{104}$$

where the first inequality follows from the PAC bound on $\tilde{\kappa}_n(\theta)$ and the definition of $\tilde{d}_n$, and the last inequality follows from the original guarantee when $\kappa_n(\theta)$ is exactly computed, and the expected revenue of the mediator (recall (9)) is

$$\sum_{n \in \mathcal{N}} (\tilde{\kappa}_n(\theta) - \tilde{d}_n) - (N-1)\,\mathbb{E}[w^\star(t)] = (N-1)\,(\tilde{\lambda} + \varepsilon''') - (N-1)\,\mathbb{E}[w^\star(t)] \tag{105}$$

$$\geq \rho + (N-1)\,(\varepsilon''' - \varepsilon''), \tag{106}$$

where the equality follows from the definition of $\tilde{d}$, and the inequality follows from the PAC bound on $\tilde{\lambda}$. $\qquad\square$

*Proof of Theorem 2.* With the choice of the parameters in the theorem, we have

$$\theta - (\varepsilon' - \varepsilon''') = \theta \tag{107}$$

$$(\rho - (N - 1)(\varepsilon'' - \varepsilon'''')) = \rho \tag{108}$$

$$(1 - \delta')^{1/(N+1)} = 1 - \delta. \tag{109}$$

Hence, Lemma 6 guarantees that the constant pivot rule $h_n(t_{-n}) = \tilde{\kappa}_n(\theta) - \tilde{d}_n$ satisfies DSIC, DE, $\theta$-IR, and $\rho$-WBB with probability $1 - \delta$. $\qquad\square$

*Proof of Proposition 3.* The constant pivot rule in Theorem 2 can be learned with $N$ independent runs of an $(\varepsilon, \delta')$-PAC BME and a single run of an $(\varepsilon, \delta')$-PAC estimator for an expectation, whose overall sample complexity is $O(N(K/\varepsilon^2)\log(1/\delta'))$. The proposition can then be established by substituting $\delta' = 1 - (1 - \delta)^{1/(N+1)}$. $\qquad\square$

### B.3  PROOF OF THE PROPOSITIONS IN SECTION 7

*Proof of Proposition 5.* Let $\mu_\star := \max_{k \in [1,K]} \mu_k$ be the best mean, $\hat{\mu}$ be the best mean estimated by Algorithm 4, and $\hat{\mu}_k^{(t)}$ be the average of the first $t$ samples from arm $k$. Let $\alpha_t := \sqrt{\frac{1}{2t} \log\left(\frac{\pi^2 K t^2}{3\delta}\right)}$. Then we have

$$\Pr(|\hat{\mu} - \mu_\star| \le \varepsilon)$$
$$\ge \Pr(\text{At every iteration, the error in the estimated mean is less than } \alpha_t \text{ for any arm in } \mathcal{R}.^3) \tag{110}$$

$$= \Pr\left(\bigcap_{t=1}^\infty \bigcap_{k \in [1,K]} \{|\{\hat{\mu}_k^{(t)} - \mu_k| < \alpha_t\}\right) \tag{111}$$

$$= 1 - \Pr\left(\bigcup_{t=1}^\infty \bigcup_{k \in [1,K]} \{|\{\hat{\mu}_k^{(t)} - \mu_k| \ge \alpha_t\}\right) \tag{112}$$

$$\ge 1 - \sum_{t=1}^\infty \sum_{k \in [1,K]} \Pr\left(\{|\{\hat{\mu}_k^{(t)} - \mu_k| \ge \alpha_t\}\right) \quad \text{by union bound} \tag{113}$$

$$\ge 1 - 2K \sum_{t=1}^\infty \exp\left(-2\alpha_t^2 t\right) \quad \text{by Hoeffding's inequality} \tag{114}$$

$$= 1 - \delta \frac{6}{\pi^2} \sum_{t=1}^\infty \frac{1}{t^2} \quad \text{by definition of } \alpha_t \tag{115}$$

$$= 1 - \delta. \tag{116}$$

$\qquad\square$

---

[3]This condition suffices because it ensures that the best arms always remain in $\mathcal{R}$.

*Proof of Proposition 6.* Let $\alpha_t := \sqrt{\log\left(\frac{\pi^2 K t^2}{6\delta}\right)/(2t)}$. Let $\mathcal{B}$ be the set of the (strictly) best arms. Let $\hat{\mu}_k^{(t)}$ be the average of the first $t$ samples from arm $k$. Then we have

$\Pr(\text{Algorithm 5 selects an } \varepsilon\text{-best arm.})$

$\geq \Pr(\text{At every iteration, all arms in } \mathcal{B} \text{ remain in } \mathcal{R} \text{ and any arm in } \mathcal{R} \text{ is } 2\alpha_t\text{-best.}^{[4]})$ (117)

$$\geq \Pr\left(\bigcap_{t=1}^{\infty} \bigcap_{k\in\mathcal{B}} \{\hat{\mu}_k^{(t)} > \mu_k - \alpha_t\} \bigcap_{\ell\notin\mathcal{B}} \{\hat{\mu}_\ell^{(t)} < \mu_\ell + \alpha_t\}\right) \tag{118}$$

$$= 1 - \Pr\left(\bigcup_{i=1}^{\infty} \bigcup_{k\in\mathcal{B}} \{\hat{\mu}_k^{(t)} \leq \mu_k - \alpha_t\} \bigcup_{\ell\notin\mathcal{B}} \{\hat{\mu}_\ell^{(t)} \geq \mu_\ell + \alpha_t\}\right) \tag{119}$$

$$\geq 1 - \sum_{t=1}^{\infty} \left(\sum_{k\in\mathcal{B}} \Pr(\hat{\mu}_k^{(t)} \leq \mu_k - \alpha_t) + \sum_{\ell\notin\mathcal{B}} \Pr(\hat{\mu}_\ell^{(t)} \geq \mu_\ell + \alpha_t)\right) \quad \text{by union bound} \tag{120}$$

$$\geq 1 - \sum_{t=1}^{\infty} K \exp(-2\alpha_t^2 t) \quad \text{by Hoeffding's inequality} \tag{121}$$

$$= 1 - \delta \frac{6}{\pi^2} \sum_{t=1}^{\infty} \frac{1}{t^2} \quad \text{by the definition of } \alpha_t \tag{122}$$

$$= 1 - \delta. \tag{123}$$

$\square$

---

[4]This condition sufficies because the algorithm stops either when $|\mathcal{R}| = 1$ or when $\alpha_t \leq \varepsilon/2$, which implies that only $\varepsilon$-best arms are in $\mathcal{R}$ when the algorithm stops.

