# OpenReview forum: "Mechanism design with multi-armed bandit"
_ICLR.cc/2025/Conference — Submitted to ICLR 2025_

### Official Review · Reviewer_ZoNd · 2024-11-03

**Soundness:** 3
**Presentation:** 2
**Contribution:** 2
**Rating:** 5
**Confidence:** 2

**Summary:**

This paper studies how to solve an LP for mechanism design. It first formulates this LP, which can satisfy four conditions, and then illustrates that the solution of this LP enjoys an exponentially smaller variable size. Then, to approximate the solution, the paper proposes to use the MAB algorithm and shows that this approximation is asymptotic optimal. Numerical simulations are also reported.

**Strengths:**

1. The numerical simulation section is designed to verify several theoretical results, which are good paper complements.

**Weaknesses:**

1. Unclear contribution. Although the paper provides an approach with computational efficiency, the LP studied in this paper differs from and looks more accessible than the prior work (Osogami et al., 2023). So, it is hard to evaluate this paper's contribution from the aspects of significance and methodology. It would be helpful if the author could discuss the technical challenges they encountered in this paper.
2. The theoretical results' organization is not easy to follow. This is a theoretical paper, providing a lot of lemmas and corollaries in Sections 5 and 6, where the essential parts are. However, the authors should put more effort into revising the presentations in these two sections. For example, in Section 5, the Lemmas 1 and 2 composes the Corollary 1. Why not directly give Corollary 1 and move Lemmas 1 and 2 to the appendix? This could help the reader quickly understand the meat of this paper.
Another example is that Corollaries 3, 4, and 5 are all on different conditions; why not just have one corollary with three bullets? For Section 6, Lemmas 4 and 5 are components to support Theorem 1. Why not use a proof sketch to posit Lemmas 4 and 5 so that readers familiar with these materials can directly skip them?

**Questions:**

### Minor Comments

- Line 170, notation $\mathcal N=[1,N]$ is confusing; how about $\{1,2,\dots,N\}$?

---

> ### Author Response · Authors · 2024-11-14
>
> > Although the paper provides an approach with computational efficiency, the LP studied in this paper differs from and looks more accessible than the prior work (Osogami et al., 2023).
>
> The LP studied in Osogami et al. (2023) is a special case of the LP studied in this paper.  Specifically, letting $\theta\equiv 0$ and $\rho=0$ in our LP gives the LP in Osogami et al. (2023).  Notice that even the LP in Osogami et al. (2023) is unlikely to admit analytical solutions in general, and this is the key technical challenge.  Our Lemma 1 gives a sufficient condition that allows us to analytically solve the LP under consideration.  Lemma 2 shows that this sufficient condition is necessary whenever types are independent, so our analytical solution is optimal for a wide range of interesting cases in mechanism design.
>
> > The theoretical results' organization is not easy to follow.
>
> We appreciate your suggestion.  We agree that the presentation can be improved.  However, some of the intermediate results are of independent interest and are worth being formally stated.  For example, Lemma 1 contains essential information that is not contained in Corollary 1.

---

> > ### Comment · Reviewer_ZoNd · 2024-11-25
> >
> > The reviewer thanks the authors for their response. The reviewer has to further questions and will keep their evaluation for now.

---

### Official Review · Reviewer_q7CJ · 2024-11-04

**Soundness:** 3
**Presentation:** 3
**Contribution:** 3
**Rating:** 6
**Confidence:** 2

**Summary:**

This work studies automated mechanism design. First, a class of optimal solutions is derived that requires an exponentially smaller number of essential variables than the previous version of linear programming. To resolve the computational issue, a connection is drawn towards best mean reward identification in MAB. Then, provably efficient design to perform best mean reward identification is provided, which is further plugged back in the original mechanism design problem.

**Strengths:**

- The automated mechanism design is an interesting problem. While I do not have exact background in this direction, I believe the efforts provided in this work are of relevance and importance to the community.

- The connection from mechanism design to multi-armed bandits is inspiring. With my background in MAB, I largely appreciate such intersection that leverages MAB techniques to faciliate other domains.

- The overall presentation and writing is clear. It has been a smooth reviewing experience for me.

**Weaknesses:**

- As I do not have a strong background in mechanism design, I would leave the further judgement of the significance and novelty of this part to other reviewers.

- For the MAB part, while the connection is interesting, I found the adopted technique is a bit straightforward. In particular, while best mean identification (BMI) and best arm identification (BAI) have their differences (e.g., the example in line 380), the upper bound is obtained in Theorem 1 is from an algorithm that perform BAI first while following up with additional samples to do BMI. I, in general, have doubts that this can be done in a more efficient way.

**Questions:**

- I would love to hear the author's opinion on the novelty of the BME design in this work. I understand that it serves as a tool for the overall mechanism design; thus it is acceptable if the novelty of this part is limited (in that case, I might need to rely on other reviewers to get an assessment for the novelty in mechanism design).

---

> ### Author Response · Authors · 2024-11-14
>
> Thank you very much for your review.
>
> > I would love to hear the author's opinion on the novelty of the BME design in this work.
>
> BME has not been studied as a problem of MAB (i.e. from the perspective of sample complexity), although related problem of estimating the best mean from given sample (i.e. bias correction) has been widely studied in machine learning (as we discuss in Line 161).  As is acknowledged in your review, a key novelty is in the connection from mechanism design to multi-armed bandits, where we reveal that the sample complexity of BME is a relevant problem.  Although we only discuss its relevance to mechanism design, BME is a fundamental problem that may find a wide range of applications, such as those that require estimating the worst case expected cost (Worst Mean Estimation).  For this new problem of BME, we show that a simple approach can achieve the best possible sample complexity.  Although the approach is simple, it is nontrivial that this simple approach matches the lower bound (as we discuss in paragraphs staring at 154, 365, and 404).  As you suggest in your review, there may be more efficient approaches that improve constant factors, have better instance-dependent complexity, etc., and we expect that our results will provide a basis for such extensions.

---

### Official Review · Reviewer_X9qJ · 2024-11-07

**Soundness:** 2
**Presentation:** 3
**Contribution:** 1
**Rating:** 3
**Confidence:** 3

**Summary:**

The paper studies mechanism design problem under multi-armed bandit framework. The authors analytically derive a class of optimal solutions for such an LP that gives mechanisms achieving standard properties of efficiency, incentive compatibility, strong budget balance (SBB), and individual rationality (IR), where SBB and IR are satisfied in expectation.

**Strengths:**

1. The paper is well written and the theoretical results appear to be correct.
2. The paper improves the previous results in Osogami [2023].
3. The paper proposes numerical experiments to show the advantages of their designs.


Osogami [2023]: Takayuki Osogami, Segev Wasserkrug, and Elisheva S. Shamash. Learning efficient truthful mechanisms for trading networks.

**Weaknesses:**

1. I hold reservations about the contributions in the paper. In Section 3, the authors introduce four properties that the mechanism needs to satisfy: Dominant Strategy Incentive Compatibility (DSIC),  Decision Efficiency (DE), $\theta$-IR, and $\beta$-WBB/SBB. Such properties should be the key challenges in the mechanism design. However, as the authors stated, directly using the VCG mechanism can satisfy the first two properties. Furthermore, regarding the other two properties, they can be represented as two linear constraints of the optimization problem. In this regard, in Section 5, the authors are essentially stating the fact "LP has a solution only when the feasible region of the constraints is non-empty", which is really trivial. In summary, I am not convinced that the method proposed in this paper is innovative or makes sense.

2. Similar to the first point, the method described by the authors in the Section 6 is essentially just the basic mean estimation of each arm's reward in stochastic MAB. The novelty of the proposed method should be further clarified.

3. The title of the paper is "Mechanism design with multi-armed bandit". However, in Section 3, the authors do not introduce any information regarding MAB.

4. In Section 3, the authors assume that the types are generated from a fixed distribution. However, in Section 6, the authors state that the algorithm can access to an arbitrary size of the sample that is independent and identically distributed (i.i.d.) according to $P(\cdots|t_n)$ for any $t_n$. These two statements seem to conflict.

**Questions:**

1. See weakness.

2. Prior to line 346, the paper does not mention MAB at all. Are the authors assuming that $ t_n \in [K] $ for all $n\in [N]$ here?

---

> ### Author Response · Authors · 2024-11-14
>
> Thank you very much for your review.
>
> > [W1] In this regard, in Section 5, the authors are essentially stating the fact "LP has a solution only when the feasible region of the constraints is non-empty", which is really trivial.
>
> Although it is trivial that every LP has a solution only when the feasible region of the constraints is non-empty, this does not mean that one can analytically derive optimal solutions to all LPs.  Our Lemma 1 gives a sufficient condition that allows us to analytically solve the LP under consideration.  Lemma 2 shows that this sufficient condition is necessary whenever types are independent, so our analytical solution is optimal for a wide range of interesting cases in mechanism design.  Note also that the (special case of) LP under consideration has been solved numerically in the prior work of mechanism design (e.g. Osogami [2023]), which also indicates the nontriviality of our results in Section 5.
>
> > [W2] Similar to the first point, the method described by the authors in the Section 6 is essentially just the basic mean estimation of each arm's reward in stochastic MAB.
>
> The suggested approach of estimating the mean reward of each arm with $O((1/\varepsilon^2) \log(1/\delta))$ samples can only guarantee that the best mean is estimated within error \varepsilon with probability at least $(1-\delta)^K$.  To provide a $(\varepsilon,\delta)$-PAC guarantee, one would need $\Omega((1/\varepsilon^2) \log(K/\delta))$ samples from each arm, resulting in the suboptimal sample complexity of $\Omega((K/\varepsilon^2) \log(K/\delta))$.  The novelty of our MAB results lies in proving that $O((K/\varepsilon^2) \log(1/\delta))$ samples are sufficient for best mean estimation, and that this is the best possible sample complexity (Theorem 1).
>
> > [W3] However, in Section 3, the authors do not introduce any information regarding MAB.
>
> The background on MAB is not needed until Section 6, and we are concerned that some of the readers would not remember what has been stated in Section 3 when they read Section 6.  However, we would appreciate the reviewer's guidance on what specific information about MAB would be most helpful to include in Section 3.
>
> > [W4] These two statements seem to conflict.
>
> The two statements do not conflict.  As is stated in Section 3, the types ($t_1, t_2, ..., t_N$) are generated from a fixed distribution $P$.  For this fixed distribution $P$, we consider a conditional distribution $P(\cdot\mid t_n)$ and take i.i.d. sample from $P(\cdot\mid t_n)$ in Section 6.
>
> > [Q2] Prior to line 346, the paper does not mention MAB at all. Are the authors assuming that $t_n\in[K]$ for all $n\in[N]$ here?
>
> Yes.  Note that we assume a finite number of players $N$ and a finite size of each type space $K$ (line 185).  In Section 6, we state our results on general MAB problems (using the language of MAB), which are then connected to mechanism design at the end of Section 6 (Theorem 2 and Proposition 3).  Specifically, for each player $n\in[N]$, the type space $\mathcal{T}_n$ corresponds to the set of arms $[K]$.

---

### Meta-Review · Area_Chair_gREJ · 2024-12-20

**Metareview:**

Various concerns about the significance of the work were raised. Reviewer q7CJ highlights that the MAB part of the paper is, in their opinion, not significant while reviewer X9qJ has concerns about the mechanism design problem and, in particular, about the significance of the studied LP. I had a closer look at the concerns and at the rebuttal and I believe that the reviewers have a point and the paper needs a major revision before being publishable.

**Additional Comments On Reviewer Discussion:**

The feedback addressed some of the reviewer concerns. The reviewers themselves were not interacting even after I posted (twice) to encourage reviewer engagement.

---

### Decision · Program_Chairs · 2025-01-22

Reject